

# Subaqueous speleothems (Hells Bells) formed by the interplay of pelagic redoxcline biogeochemistry and specific hydraulic conditions in the El Zapote sinkhole, Yucatán Peninsula, Mexico

Simon Michael Ritter[a, *], Margot Isenbeck-Schröter[a, b], Christian Scholz[a], Frank Keppler[a, b], Johannes Gescher[c, d], Lukas Klose[a], Nils Schorndorf[a], Jerónimo Avilés Olguín[e], Arturo González-González[f] and Wolfgang Stinnesbeck[a, b]

[a] Institute of Earth Sciences, Heidelberg University, Im Neuenheimer Feld 234-236, D-69120 Heidelberg, Germany
[b] Heidelberg Center for the Environment (HCE), Heidelberg University, D-69120 Heidelberg, Germany
[c] Institute for Applied Biosciences, Department of Applied Biology, Karlsruhe Institute of Technology (KIT), Karlsruhe, Germany
[d] Institute for Biological Interfaces, Karlsruhe Institute of Technology (KIT), Eggenstein-Leopoldshafen, Germany
[e] Instituto de la Prehistoria de América, Carretera federal 307, km 282, Solidaridad, 77711 Solidaridad, Quintana Roo, México
[f] Museo del Desierto, Carlos Abedrop Davila 3745, Nuevo Centro Metropolitano de Saltillo, 25022 Saltillo, Coahuila, Mexico

*Corresponding author: S. Ritter, simon.ritter@geow.uni-heidelberg.de

## Abstract

Unique bell-shaped underwater speleothems were recently reported from the deep (~55 m) meromictic El Zapote sinkhole (cenote) on the Yucatán Peninsula, Mexico. The local diving community has termed these speleothems as Hells Bells because of their shape and appearance in a lightless environment in ~28–38 m water depth above a sulfidic halocline. It was also suggested that Hells Bells form under water, yet the mystery of their formation remained unresolved. Therefore, we conducted detailed hydrogeochemical and geochemical analyses of the water column and Hells Bells speleothems including stable carbon isotopes. Based on the comprehensive results presented in this study we deduce that both, biogeochemical processes in the pelagic redoxcline and a dynamic halocline elevation of El Zapote cenote, are essential for Hells Bells formation. Hells Bells most likely form in the redoxcline, a narrow 1–2 m thick water layer immediately above the halocline where a pelagic chemolithoautotrophic microbial community thrives from the upward diffusion of reduced carbon, nitrogen and sulfur species released from organic matter degradation in organic-rich debris. We hypothesize that chemolithoautotrophy, in particular the proton consuming nitrate-driven anaerobic sulfide oxidation, favors calcite precipitation in the redoxcline and hence Hells Bells formation. A dynamic elevation of the halocline as a hydraulic response to recharge events, e.g. hurricanes, is further discussed, which might explain the shape of Hells Bells as well as their occurrence over a range of 10 m water depth. Finally, we infer apparent prerequisites for Hells Bells formation considering the exclusivity of these underwater speleothems to only a few cenotes of a restricted area of the northeastern Yucatán Peninsula.





## 1. Introduction

Pending speleothems, also called stalactites or dripstones, result from physicochemical processes under subaerial conditions in a cave atmosphere. Calcite precipitates due to $CO_2$-degassing and evaporation of water enriched in dissolved carbonate dripping into the cave. Normally, stalactites rejuvenate and form a tip at the lower end from which the water drips to the cave

floor. Nevertheless, in recent years, researchers have identified a small group of stalactites that appear to have calcified underwater. For these formations, interactions between physicochemical and biological calcite precipitation processes are interpreted (Barton and Northup, 2007; Bontognali et al., 2016; Gradzinski et al., 2012; Guido et al., 2013; Holmes et al., 2001; Jones et al., 2008, 2012; Macalady et al., 2007; Macintyre, 1984; Melim et al., 2001; Queen and Melim, 2006). We recently presented a spectacular example for these subaqueous speleothems termed Hells Bells from El Zapote sinkhole

about 26 km west of Puerto Morelos on the Yucatán Península of southern Mexico (Fig. 1) (Stinnesbeck et al., 2017b). These bell-shaped structures consist of calcite and reach lengths of up to two meter. Hells Bells conically expand downward with strictly horizontal ring-like concentric swellings and neckings on the surface (Fig. 2). Apparently, they form in a lightless environment in freshwater above the anoxic and sulfidic halocline (Stinnesbeck et al., 2017b). Because of these toxic environmental conditions in complete darkness, the local diving community has termed the El Zapote speleothem

formations as Hells Bells. They grow from the cavern ceiling and wall. Additionally, small individuals also cover a tree that has fallen into the sinkhole around ~3.5 cal kyr BP, which indicates that Hells Bells must have formed during the Holocene to at least historical times, and thus at periods when the deep sections of the cenote had already been submerged for thousands of years (Stinnesbeck et al., 2017b). Thus, the conditions for the formation of the biggest underwater speleothems worldwide must have existed consistently throughout the past thousands of years at El Zapote cenote.

The internal structure of Hells Bells is characterized by laminar fabrics of alternating units of elongated dogtooth spar calcite and microcrystalline spar calcite. Microspar layers and corroded lobes of dogtooth spar crystals indicate either discontinuous growth of Hells Bells and/or intermittent dissolution. Phylogenetic analyses of Hells Bells speleothem surfaces from specimens of 33 and 34 m water depth indicate that microorganisms inhabiting the Hells Bells potentially support a full nitrogen-circle and autotrophic growth (Stinnesbeck et al., 2017b). The growth of Hells Bells may thus be mediated by

specific physical and biogeochemical conditions above and in the halocline, while formation of Hells Bells was likely restricted to the lowermost part of the freshwater body. However, due to the limited available data including geochemical parameters, the suggested processes for Hells Bells formation were regarded as highly speculative.

Therefore, in this study we conducted detailed geochemical analysis including stable carbon isotopes of the water body and Hells Bells speleothems of El Zapote cenote. Based on the results we present a hypothesis on the subaqueous growth of these

exceptional structures. We deduce that both, biogeochemical processes in the pelagic redoxcline and a dynamic halocline elevation of El Zapote cenote, are essential for Hells Bells formation.



## 1.1 Geological background

The northeastern Yucatán Peninsula consists of horizontally layered shallow-water carbonates of Mio-, Plio- and Pleistocene ages (Lefticariu et al., 2006; Weidie, 1985) and probably hosts the largest network of underwater caves in the world. The Mexican state of Quintana Roo alone counts more than 370 underwater caves, with a total estimated length of >7000 km and

a confirmed total length of ~1460 km and individual cave systems reaching up to >350 km in length (QRSS, 2018). These karst cave systems developed predominantly by the interaction of glacioeustacy, littoral processes and mixing-zone hydrology during glacial periods of the Pleistocene (Smart et al., 2006; Weidie, 1985). Precipitation rapidly infiltrates through the porous limestone into the underlying coastal aquifer consisting of a meteoric water mass, the fresh water lens above a marine water mass intruding from the coast (e.g. Kovacs et al., 2017a). The thickness of the freshwater lens varies

between ~10–100 m and is generally thinner towards the coast (Beddows et al., 2002), while seawater intrudes up to 100 km inland (Beddows et al., 2007). The halocline separates the meteoric and marine water bodies and is usually characterized by undersaturation with respect to $CaCO_3$, leading to cave formation and conduit enlargement in the coastal carbonate aquifer (Back et al., 1986; Gulley et al., 2016; Mylroie and Carew, 1990; Smart et al., 2006). The depth of the halocline increases with distance from the coast (Bauer-Gottwein et al., 2011); areas closer to the coast show a higher salinity of the freshwater

lens than inland areas (Kovacs et al., 2017b). The position of the halocline is also dependent on global sea level and the thickness of the freshwater lens. Hydraulic gradients are generally very low with values of 1–10 cm $km^{-1}$ (Bauer-Gottwein et al., 2011 and references therein). Although Moore et al. (1992) and Stoessell et al. (1993) report that the thickness of the freshwater lens does not vary significantly between seasons or on a yearly basis, Escolero et al. (2007) documented a significant halocline elevation of up to 17.5 m in response to great recharge events, such as hurricanes.

Sinkholes (locally called *cenotes*) were formed by dissolution and collapse of the carbonate rock. They are common throughout the Yucatán Peninsula, connecting the subterranean cave system with the surface (Bauer-Gottwein et al., 2011). For more detailed information about the formation and occurrence of cenotes on the Yucatán Peninsula we refer the readers to Torres-Talamente et al. (2011) and Schmitter-Soto et al. (2002).

## 1.2 El Zapote cenote

El Zapote cenote is located 26 km west of Puerto Morelos on the Yucatán Peninsula of southeast Mexico (20°51´27.78" N 87°07´35.93" W; Fig. 1). In cross-section the cenote is bottle-shaped with a deep vertical water-filled shaft that opens at 28 m water depth to a wide cavern of 60 to >100 m in diameter that reaches to about 54 m water depth with a 20 m high debris mount in the center (Fig. 3 a). A fallen tree stands on top of the debris mount and small Hells Bells cover the stem. There are no apparent passages or conduits that connect El Zapote cenote to a cave system. Additional details on El Zapote cenote are

given in Stinnesbeck et al. (2017b) and in Stinnesbeck et al. (2017a) who described the new genus and species of a giant ground sloth, *Xibalbaonyx oviceps,* from an individual that was found on the floor of El Zapote cenote.



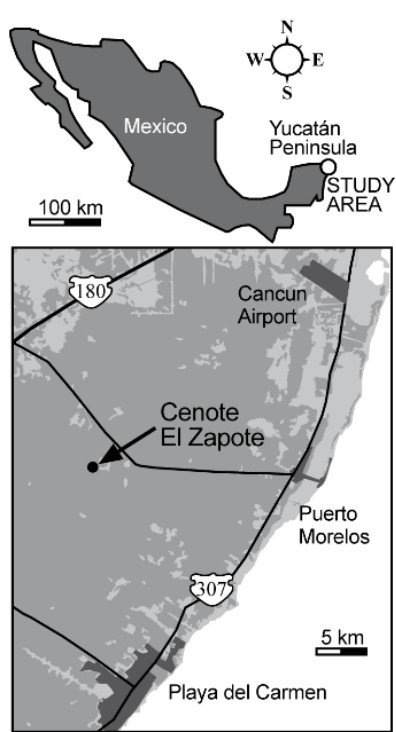

**Fig. 1: Location of the studied El Zapote cenote with respect to Mexico and the Yucatán Peninsula.**







Fig. 2: Technical diver in El Zapote cenote during a sample collecting dive carrying the Niskin bottle, sample containers and the multiparameter sonde attached to the sidemount gas bottle (a). Turbid layer immediately above the halocline forming a distinct horizontal white cloud at around 36 m water depth (b). Transition of cenote shaft to the open dome-shaped cavern at 28 m water depth (c), where the vertical wall of the cenote shaft is devoid of Hells Bells speleothems (upper part of picture c), whereas small



specimens of Hells Bells grow down from the horizontal ceiling below (lower part of c). Brown-colored Mn-oxide coatings on host rock carbonates and Hells Bells speleothems reach down from 28 m water depth to around 30 m water depth at the transition of the cenote shaft to the wide dome-like cavern (c and d). Below around 30 m water depth, Hells Bells speleothem and host rock carbonate surfaces are devoid of brownish Mn-oxide coatings. They are white to light-grey colored revealing a distinct horizontal
boundary (d). Close-up shot of the lowermost calcite rim of a Hells Bell speleothem at around 32–35 m water depth showing mm-sized calcite crystals (e).

## 2. Methods

### 2.1 Sampling

Sampling at El Zapote Cenote was carried out between December 10[th] and 15[th] 2017. Water samples were taken early in the
morning prior to any touristic diving group activities to ensure sampling of an undisturbed water column. Water sample recovery and the recording of the in-situ parameters were carried out with a winch from the surface down to the top of the debris mount (0–36 m) and by technical divers from the top of the debris mount down to the cenote floor following the slope of the debris mount (Fig. 2 a).

In-situ parameters pH ($\pm$0.1), eH ($\pm$20 mV), dissolved oxygen ($\pm$0.1 mg l$^{-1}$, detection limit 0.2 mg/l), electrical conductivity
($\pm$0.05 % of value), temperature ($\pm$ 0.01 °C) and turbidity ($\pm$2 % of value) were determined with a multiparameter water sonde EXO-1 (Xylem Analytics, Norway). All parameters, including water depth via pressure measurement, were internally logged by the sonde ($\pm$0.04 m). Water depths were corrected to the ambient air pressure of the respective day of sampling. In order to reach the greatest possible water depth, a total of four winch-operated profiles were run within 2 days, with laterally shifting starting points of the profile at the surface. In order to complete the measurement in the whole water column,
technical divers carried the EXO-1 Probe with them during sampling. Due to increasing sulfide concentrations in water depths below the turbid layer and interaction of sulfide with the Ag/Cl pH electrode, a shift of pH of up to 0.2 pH units towards higher values was observed when comparing the pH logs of the way down with the pH logs of the way up (Fig. S1). This shift is dependent on the exposure time of the electrode and the respective sulfide concentrations and could not be quantified nor corrected for. However, the sensor recovers to initial pH values after a certain time in non-sulfide water.
Therefore, the pH values presented in this study are representative for the water column from 0 to 37 m water depth and are overestimated in water depths from 37–50 m where the actual absolute pH values are most likely lower, i.e. more acidic.

Water samples from 15–35 m water depth were retrieved using a winch and a 5 L polyethylene Niskin bottle (Hydrobios, Kiel, Germany). Sampling depths represent the center of the 0.6 m tall sampling bottle and were determined by cable length with a depth counter attached to the winch. Water samples from 35.2–45 m water depth were retrieved by technical divers
(Fig. 2 a). Water samples collected by the technical divers were taken with 120 ml PE-containers and the water depth was noted for each sample. Water samples for the analysis of dissolved gases ($CO_2$, $CH_4$) were taken in 24 ml glass vials and sealed underwater at the respective depth (four samples at each depth level). The EXO-1 sonde was attached to a side mounted compressed air bottle pointing towards the front of the technical diver in order to record the in-situ parameters of each water sample (Fig. 2 a). The depth of the water samples taken by technical divers was corrected to the depth of the





attached logging device (EXO-1). For four samples between 35 and 37 m, depth was interpreted from the increase of sodium and chloride contents correlated to the electrical conductivity in this interval.

Water samples were treated on-site immediately after the water samples were retrieved. Samples for determination of dissolved ions were taken with 20 ml sterile polypropylene syringes and then filtered through a cellulose acetate filter (0.45 µm). Samples for cation determination were acidified with 50 µl of 65 % $HNO_3$ (A.G.) to adjust a pH <2; they were stored in 15 ml Falcon polypropylene centrifuge tubes. Samples for anion determination were taken equivalent, but not acidified, and stored cool in 15 ml Falcon polypropylene centrifuge tubes. Samples for the determination of dissolved inorganic carbon (DIC) and dissolved organic carbon (DOC) were filtered through a cellulose acetate filter (0.45 µm), stored in 24 ml glass vials and sealed gas-tight. Samples for the determination of content and isotopic ratios of the dissolved gases $CH_4$ and $CO_2$ were filled in 24 ml glass vials; subsequently 100 µl 60 % $HgCl_2$ solution was added via a syringe pierced through the septum to sterilize the samples.

A large volume sample (5 l) of the turbid layer water at around 36 m water depth was taken with a Niskin bottle by technical divers and subsequently filtered through a 0.45 µm cellulose acetate filter with a vacuum pump. The filter was air-dried; back in the laboratory a small piece of the filter was coated with carbon for subsequent secondary electron (SE) imaging and analyses.

Sulfide and nitrite were determined on-site by photometric analysis with a Photometer (Hach Lange DR200) using Merck Spectroquant spectrometric methods

Technical divers collected several Hells Bells samples grown on the tree trunk from 7 water depth levels between 32.7 and 37.3 m. To obtain the youngest part of individual Hells Bells growth, samples were first studied under the microscope. Only samples with apparently fresh, well accentuated crystal tips were chosen for analysis geochemical and stable isotope analysis.

## 2.2 Analytical measurements

### 2.2.1 Major and trace element analysis

Major cation concentrations (Ca, Mg, Sr, Ba, K, Na, Si) of water samples were determined by optical emission spectroscopy with an Agilent 720 ICP-OES. Quality control was performed using the reference materials SPS-SW1, SPS-SW2 and TMDA 70.2. Recovery rates were in the range of 97–104 % for the analyzed elements. Measurement precision for each element was <2 % (RSD, n=3).

Concentrations of anions $Cl^-$, $SO_4^{2-}$ and $NO_3^-$ were determined with ion-chromatography (Dionex ICS-1100) with a RSD of <3 % derived from long-term repeated analysis of reference material SPS-WW1 NUTR. The concentrations of DOC were determined with a Total Carbon Analyzer (Shimadzu TOC-CPH) with a RSD of <2 % derived from repeated analysis of an in-house standard water.





Orthophosphate was determined by the photometric molybdenium phenyl blue method on 880 nm light extinction with a UV/VIS photometer (Specord 50, Analytic Jena).

Trace element concentrations (Fe, Mn, Mo, P, S, U) of selected water samples were determined by HR-ICP-MS (Thermo Finnigan Element 2). Analyses were normalized by an internal Indium-standard. Calibration solutions were prepared with the MERCK VI Multi Element Standard. The recovery rates of SLRS5 reference material were 95 % (Fe), 93 % (Mn), 82 % (Mo) and 133 % (U) with respect to the referenced values and P and S were within the range of reported uncertified values. The precision was <3.7 % (RSD) derived from repeated (n=5) measurements of the reference material in the measurement run.

Around 3 mg of the powdered speleothem samples were digested in 2ml 10%$HNO_3$ for major and trace cation analyses. Subsequently, concentrations of Ca, Mg, Sr, Ba, P, S, Fe and Mn of the aliquots were determined by ICP-OES. Quality control of the measurement was performed using reference materials SPS-SW1 and SPS-SW2 with recovery rates ranging from 99 to 111 % for the analyzed elements. Quality control for digestion of the carbonate material was performed with limestone reference material ECRM 752-1 with recovery rates between 106–110 % for the elements Ca, Mg, Ba, Sr and Mn and 82 % for the element Fe.

Calcite saturation and $HS^-$ activity was calculated with PhreeqC (Parkhurst and Appelo, 1999) using phreeqc.dat. The diffusion J was calculated with the first Fick´s law with diffusion coefficients of $D_{O_2}$, $D_{NO_3^-}$ and $D_{HS^-}$ of 2.1, 1.9 and 1.2 $10^{-9}$ $m^{-2} s^{-1}$, respectively taken form phreeqc.dat (Parkhurst and Appelo, 1999).

### 2.2.2 Stable carbon isotope and concentration measurements of $CH_4$ and $CO_2$

For the determination of dissolved gases, a 5 ml headspace with nitrogen gas ($N_2$ 99.999 %) was created in each of the four samples of the respective water depth. Samples were taken for the analysis of dissolved gases at ambient laboratory temperatures of 23 °C. After equilibration (~24h), the headspace of the four samples was transferred and combined in one 12 ml evacuated exetainer vial. To ensure a pressureless transfer of the gas phase from the headspace to the exetainer, a brine solution of 200 g $l^{-1}$ NaCl was introduced at the bottom of the vial and the gas-phase was simultaneously removed and subsequently transferred to evacuated exetainer vials. Concentrations $CH_4$ and $CO_2$ in the gas samples were measured as follows: Headspace samples (50µl) were injected in a flow of 1mL min-1 of helium with a split ratio of 5:1 to a Shin Carbon ST column (80/100 mesh, 2m x 0.53mm i.d., Restek Corporation) quantified by a gas chromatograph (GC-2010 Plus, Shimadzu Corporation, Kyoto, Japan) coupled to a Barrier Ionization Discharge (BID) detector (BID-2010 Plus, Shimadzu Corporation, Kyoto, Japan). The GC oven was initially held at 30°C for 1 min and then ramped at 10°C/min to 200°C. Quantification of $CH_4$ and $CO_2$ was carried out by comparison of the integrals of the peaks eluting at the same retention time as that of the authentic standard with calibration curves. The dissolved concentrations of $CH_4$ in the water were then calculated from the measured mixing ratio using Henry's law (Wiesenburg and Guinasso, 1979) and solubility coefficients for $CH_4$ according to Weiss (1974) and Yamamoto et al. (1976).




Stable carbon isotope ratios of $CO_2$ ($\delta^{13}$C-$CO_2$ values) were analyzed by gas chromatography stable isotope ratio mass spectrometry (GC-IRMS) by a HP 6890N gas chromatograph, coupled to a 253 Plus™ isotope ratio mass spectrometer (ThermoQuest Finnigan, Bremen, Germany) with average analytical uncertainties of 0.2‰ for $\delta^{13}$C-$CO_2$ values. 2 $\sigma$ uncertainties were derived from 5 replicates. All $^{13}$C/$^{12}$C isotope ratios are expressed in the conventional $\delta$ notation in per mil

versus VPDB, defined in Eq (1):

$$\delta^{13}C_{V-PDB} = \left[ \left( ^{13}C/^{12}C_{sample} \right) / \left( ^{13}C/^{12}C_{standard} \right) \right] - 1 \qquad (1)$$

For details of the $\delta^{13}$C-$CO_2$ measurements by GC-IRMS we would like to refer to previous studies by Keppler et al. (2010) and Laukenmann et al. (2010).

Stable carbon isotope ratios of $CH_4$ ($\delta^{13}$C-$CH_4$ values) were determined (GC-IRMS). In brief, $CH_4$ of the sample was trapped

on Hayesep D and then transferred to the IRMS system (ThermoFinnigan Delta$^{plus}$ XL, Thermo Finnigan, Bremen, Germany). The working reference gas was carbon dioxide of high purity (carbon dioxide 4.5, Messer Griesheim, Frankfurt, Germany) with a known $\delta^{13}$C-$CH_4$ value of -23.634 ‰ ± 0.006 ‰ versus V-PDB. All $\delta^{13}$C-$CH_4$ values were corrected using two $CH_4$ working standards (isometric instruments, Victoria, Canada) and normalized by two-scale anchor calibration according to Paul et al. (2007). The average standard deviation of the analytical measurements was in the range of 0.1 ‰ to

0.3 ‰.

The $\delta^{13}$C-$HCO_3^-$ values were calculated from the measured $\delta^{13}$C-$CO_2$ of the headspace of the water samples that was generated in the laboratory as equilibrium fractionation at 23° C ($\delta^{13}$C-$CO_2$ + 8.16 ‰ = $\delta^{13}$C- $HCO_3^-$) after Mook (2000).

For stable carbon isotope analyses of carbonates, approximately 50 µg of powdered speleothem subsamples was analyzed using a ThermoFinnigan MAT253Plus gas source mass spectrometer equipped with a Thermo Fisher Scientific Kiel IV

Carbonate Device at Heidelberg University. Values are reported relative to VPDB (Eq. 1) through the analysis of an in-house standard (Solnhofen limestone) calibrated to IAEA-603. The precision of the $\delta^{13}$C analyses is better than 0.08 ‰ and 0.06 ‰ (at 1σ level), respectively.

### 2.2.3 Optical methods

Hells Bell specimen ZPT 7, described in Stinnesbeck et al. (2017b) was vertically cut in half and thin sections were prepared

from one half of the specimen. Photographs of the thin sections were taken with a Keyence VHX-6000.

Polished counterparts of the thin sections and small pieces of Hells Bells were coated with carbon for secondary electron (SE) imaging and energy dispersive X-ray (EDX) analyses. SE-imaging and element mapping was performed with a Leo 440 at 20 kV with a X-Max 80 mm$^2$ detector.





## 3. Results

### 3.1 Hydrogeochemistry

The water column of the El Zapote cenote is stratified into an oxygenized fresh water body overlying an anoxic transition zone of increasing electrical conductivity (EC), the halocline, and an anoxic salt water body below (Fig. 3 a). Water

temperatures vary little between 0 to 30 m water depth (24.37–24.42 °C); a steep increase is identified in a narrow zone from 30–32 m water depth (24.42–24.55 °C), followed by almost invariable temperatures from 32 m water depth (24.55 °C) down to the bottom of the cenote (25.22°C) (Figs. 3 a and b). A distinct density boundary, the top of the halocline, is identified at 36.6 m water depth by a steep increase in EC. Sea water-like salinity is reached at around 46 m water depth indicating a thick halocline layer of around 10 m thickness (Figs. 3 a and b). Low turbidity readings indicate clear water throughout the

water column, except for a ~1.6 m thick layer of increased turbidity immediately above the halocline from 35.0–36.6 m water depth, with a peak of 8.0 FNU detected at 35.8 m water depth (Figs. 3 a and b; Table S1). This turbid layer is also easily detected macroscopically in the water column as a white cloudy layer (Fig. 2 b) and coincides with a distinct redoxcline from ~35–37 m water depth, in which the redox potential (EH) decreases from ~250 to ~140 mV (Figs. 3 a and b). Dissolved oxygen (DO) decreases nearly linear from 30 m to concentrations below detection limit at ~35 m water depth

just above the turbid layer. Below, DO was below detection limit (Figs. 3 a and b). The pH shows neutral values from 0–30 m water depth and slightly decreases to 6.90 at the top of the turbid layer (Fig. 3 a). Within the turbid layer pH values increase to more alkaline values of around 6.94 at 35.8 m water depth. The pH values decrease again below the turbid layer to 6.73 at 40 m and invariably remain at about this value down to 48 m. From there, values increase to about neutral (6.95) close to the cenote bottom at 49 m water depth (Figs. 3 a and b).

Concentrations of the major dissolved ions $Na^+$, $Cl^-$, $Ca^{2+}$, $Mg^{2+}$ and $SO_4^{2-}$ reflect the stratification of the water column in the cenote, with generally low concentrations in the fresh water body from 0–30 m water depth and slightly increasing concentrations from 30 m water depth to the top of the turbid layer at 35 m water depth, a stronger increase within the turbid layer from 35–36.6 m water depth, and an even stronger increase from the top of the halocline at 36.6 m water depth down to the cenote bottom (Fig. 3 c and Table S2). Mg/Ca ratios strongly increase from the top of the turbid layer at 35 m water

downwards, due to higher Mg concentrations in the salt water body (Fig. 3 c). Although sulfate concentrations increase downwards from the top of the halocline, a relative decrease of $SO_4^{2-}$ ions is detected, compared to the chemically conservative ion $Cl^-$, by a decrease in $SO_4^{2-}/Cl^-$ within the turbid layer and below in the halocline (Fig. 3 c).



**Fig. 3: Hydrogeochemistry of the water column of El Zapote cenote.** The horizontal grey band indicates the depth position of the turbid layer, while the dashed line indicates the top of the halocline at 36.6 m water depth. a: Water in-situ parameters versus water depth (left) in relation to the El Zapote cenote cross section (right). In-situ parameters and samples were taken along a winch profile and a diver profile as shown in the cenote cross section. Note the logarithmic scale of the electrical conductivity (EC). b: Close-up of the water in-situ parameters in 31–41 m water depth. Note that the scale of EC is non-logarithmic and is only shown for the range between 1–5 mS cm$^{-1}$, in order to point out the increase in salinity at the beginning of the halocline. c: Water





**chemical parameters determined in the water column between 31–41 m water depth. Na⁺ and Cl⁻ concentrations are only shown in the range of 0–80 mmol l⁻¹ to highlight the concentration pattern above and within the halocline.**

Concentrations of DIC are about 7.8 mmol l$^{-1}$ in the fresh water body. They increase in the turbid layer and show a peak at 40 m water depth with concentrations increasing to 14.5 mmol l$^{-1}$; below, they decrease towards the cenote bottom (Fig. 3 c and Table S2). The dissolved organic carbon (DOC) concentrations are low in the fresh water body and show a distinct peak within the turbid layer, coinciding with the peak in turbidity at 35.7 m water depth (Fig. 3 c). Below the turbid layer DOC concentrations slightly increase and peak at 39–40 m water depth, decreasing from there towards the cenote bottom (Fig. 3 c and Table S2). Nitrate concentrations are ~50 µmol l$^{-1}$ in the fresh water unit of the cenote shaft (Table S2). They decrease from 30 m water depth towards the top of the turbid layer and rapidly fall below detectable concentrations within this layer (Fig. 3 c). Nitrite peaks in a narrow zone at the top of the turbid layer with concentrations of up to 0.8 µmol l$^{-1}$ (Fig. 3 c). High total sulfide (S(-II)) concentrations of up to 5.6 mmol l$^{-1}$ were detected in 40 m water depth. Concentrations decrease upwards, fading in the lower part of the turbid layer at 36 m water depth (Fig. 3 c). Below the 40 m depth level, S(-II) concentrations decrease to values around 3 mmol l$^{-1}$ down to 45 m water depth (Fig. 3 c and Table S2). Concentrations of dissolved CH$_4$ (CH$_4$(aq)) are low in the fresh water body with values of about 0.09 µmol l$^{-1}$. Methane concentrations increase from the turbid layer at 36 m water depth downwards to values of 25 µmol l$^{-1}$ at 39 m water depth (Fig. 3 c).

### 3.1.1 Calcite Saturation

The calculated saturation index (SI) of calcite shows calcite saturation in the fresh water body and the uppermost part of the halocline with values from 0.03–0.07 (Fig. S2). The SI closely follows the pH in the fresh water body revealing a distinct peak of slightly higher values of SI = 0.07 in the turbid layer at ~36 m water depth. However, SI values calculated for the halocline suffer from the overestimated pH readings in the extremely sulfidic water of the halocline and are therefore not considered.

### 3.1.2 Trace elements

Dissolved iron and manganese concentrations are very low in the fresh water body with concentrations of 0.1 and 0.01 µmol l$^{-1}$, respectively, and slightly increase within the turbid layer towards the salt water body, to concentrations of up to 0.47 (Fe) and 0.06 (Mn) µmol l$^{-1}$ (Fig. S3). Phosphate and silica concentrations are invariably low in the fresh water body (P$_{ortho}$ ~0.25 and Si ~63 µmol l$^{-1}$) and increase in the salt water body peaking at 40 m water depth with concentrations up to 10.3 (P$_{ortho}$) and 275 (Si) µmol l$^{-1}$ (Fig. S3). Uranium content correlates to the redox potential of the water as indicated by uniform contents of ~0.012 µmol l$^{-1}$ in the fresh water column and rapidly decreasing values in one order of magnitude in the turbid layer, to 0.0012 µmol l$^{-1}$ at 40 m water depth (Fig. 4 and Table S2).



### 3.1.3 Stable carbon isotopes of DIC and CH$_4$

The $\delta^{13}$C-HCO$_3^-$ values at water depth from X to Y are shown in Figure. 5. The average $\delta^{13}$C-HCO$_3^-$ value is -9.8 ‰ in the fresh water body where DIC content is about 8 mmol l$^{-1}$. In the turbid layer $\delta^{13}$C-HCO$_3^-$ values show a distinct peak towards less negative values of up to -7.9 ‰ at slightly increasing DIC concentrations. Below the turbid layer $\delta^{13}$C-HCO$_3^-$ values

5    rapidly decrease towards more negative values of -12.4 ‰ between 39 and 42 m water depth at increasing DIC concentrations (Fig. 4). A rather slight increase of $\delta^{13}$C-HCO$_3^-$ values (-11.6 ± 0.7 ‰) is observed towards the cenote bottom at 44 m water depth (Table S3).

The $\delta^{13}$C-CH$_4$ values are shown alongside with the CH$_4$ concentrations in Figure 5. The pattern of $\delta^{13}$C-CH$_4$ within the water column is similar to that of $\delta^{13}$C-HCO$_3^-$. In the fresh water body values of $\delta^{13}$C-CH$_4$ are approximately constant at about -49

10   ‰ and CH$_4$ concentrations are very low, roughly corresponding to that of atmospheric equilibrium (0.04–0.09 µmol l$^{-1}$). $\delta^{13}$C-CH$_4$ increases to -28 ‰ within the turbid layer and again decreases to -61 ‰ below the turbid layer, while CH$_4$ concentrations increase within and below the turbid layer (Fig. 4).

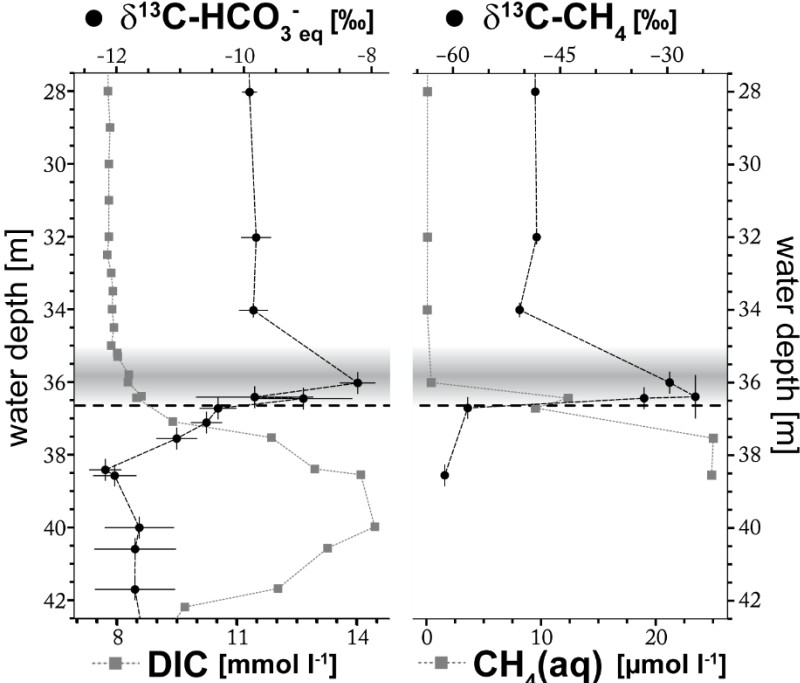

**Fig. 4: Stable carbon isotope values $\delta^{13}$C-HCO$_3^-$$_{eq}$ of the dissolved HCO$_3^-$ in equilibrium with $\delta^{13}$C-CO$_2$ values measured in**

15   **headspace and $\delta^{13}$C-CH$_4$ values of water samples alongside the concentrations of DIC and CH$_4$ of water samples. The grey band represents the turbid layer in 35–36.6 m water depth and the horizontal dashed line indicates the top of the halocline at 36.6 m water depth. Horizontal error bars represent 2σ uncertainties, vertical error bars indicate up to 0.6 m uncertainty of gas samples that where not taken from the sample used to determine chemical parameters (see section 2.1).**



## 3.2. Petrography of Hells Bells speleothems

Petrographic characteristics of Hells Bells are shown in Fig. 5. The size of individual crystals of Hells Bells is varying from µm scale to several mm-sized crystals that are easily identified macroscopically. The latter are frequently dominant in the youngest calcite rims at the bottom of Hells Bells from water depths reaching from ~28 to ~35 m (Fig. 2 e). Hells Bells from

5    greater water depths show rounded or globular calcite surfaces at the lowermost margin of the speleothems indicating dissolution (Fig. 2 f). SEM-images of the lowermost part of Hells Bell surfaces are shown in Figs. 6 a, b and c. The calcite morphology varies from bladed or book-like calcite crystals (Fig. 5 a), dogtooth-like calcite crystals (Fig. 5 b) and blocky calcite rhombs (Fig. 5 c). In thin sections of the specimen ZPT-7 (Fig. 5 d1) (see also Stinnesbeck et al., 2017b), these crystal morphologies are expressed as rather botryoidal (dog-tooth-like and bladed-shaped) and mosaic calcite phases (blocky

10   calcite rhombs). Electron images of the polished counter pieces of the thin section are shown in Fig. 5 d2. An element map of Mg shows that botryoidal calcite phases incorporated more Mg (appearing brighter in Fig. 5 d3) than the mosaic calcite phases (appearing darker in Fig. 5 d3).





Fig. 5: **Petrographic characteristics of Hells Bells speleothems. SE-images of Hells Bell samples Z17-8DC (a), Z17-18J (b), Z17-9J (c) of El Zapote cenote showing bladed (a), dogtooth-like (b) and blocky (c) calcite rhombs. Polarized transmitted light-microscopic images of a thin section from ZPT-7 (d1) (shown in Stinnesbeck et al., 2017b) showing different calcite fabrics of**



angular coarse-grained mosaic calcite (mo) and fine grained elongated botryoidal calcite (by). The same detail is shown in the BSE-image of the polished counter slab that corresponds to the thin section (d2). The Mg-element map (d3), where higher abundances of Mg appear brighter, indicates a difference in Mg content between the botryoidal and mosaic calcite phases. The white rectangles represent areas of measured integrated element spectra.

## 3.3 Geochemistry of Hells Bells speleothems

Samples were collected from the lowermost and presumably youngest part of Hells Bell specimens that grew on a ceiba tree fallen into the El Zapote cenote at about 3.5 cal kyr BP (Stinnesbeck et al., 2017b). They were analyzed for major and trace elements and stable carbon isotopes. The results are given in Table. 1.

### 3.3.1 Major and trace elements

The calcite of Hells Bell speleothems revealed no residues after digesting ~3 mg sample in 12 ml dilute 1 M Nitric indicating that Hells Bells calcite is devoid of acid soluble impurities. The Mg/Ca, Sr/Ca and Ba/Ca ratios show narrow ranges with mean values of $22.5\pm2.9*10^{-3}$, $38.6\pm5.9*10^{-5}$ and $1.10\pm0.31*10^{-5}$, respectively. They are closely related and positively correlate in each sample (Fig. 6 a). There is also a trend towards decreasing ratios with increasing water depth of the respective sample (Fig. 6 b). Iron and manganese show more variable concentrations with ratios of Fe/Ca and Mn/Ca between $3.0–11.3*10^{-5}$ and $16–39.3*10^{-6}$, respectively. Iron and manganese show a weak positive correlation but no dependency on water depth. The content of sulfur in Hells Bells carbonate is constantly high with concentrations of 0.8–1.0 g kg$^{-1}$ (Table S4) and mean S/Ca ratios of $2.87\pm0.42*10^{-3}$, showing no dependency on water depth of the sample (Table 1).

### 3.3.2 Hells Bells stable carbon isotopes

Stable carbon isotope values of Hells Bells calcite samples ($\delta^{13}C_{calcite}$) from different water depth range from -12.85 to -13.82 ‰ with a mean value of -13.37±0.70 ‰ (n=9, Table 1). There is a weak correlation of increasing $\delta^{13}C_{calcite}$ values with water depth of the samples (Fig. 6 b). Furthermore, $\delta^{13}C_{calcite}$ values show a strong positive correlation with Sr/Ca and Ba/Ca with r$^2$ of 0.82 and 0.89 (Fig. 6 a).

The stable carbon isotope ratio of the HCO$_3^-$ that is in equilibrium with the Hells Bells calcite ($\delta^{13}C\text{-}_{eq}HCO_3^-$) at 25 °C water temperature ($\delta^{13}C_{Calcite} - 0.91$ ‰ $= \delta^{13}C\text{-}_{eq}HCO_3^-$) was calculated after Mook (2000). The calculated $\delta^{13}C\text{-}_{eq}HCO_3^-$ is -14.28±0.70 ‰ which is slightly lower than the $\delta^{13}C\text{-}HCO_3^-$ determined for the water column with a range of -9.1 to -12.3 ‰.





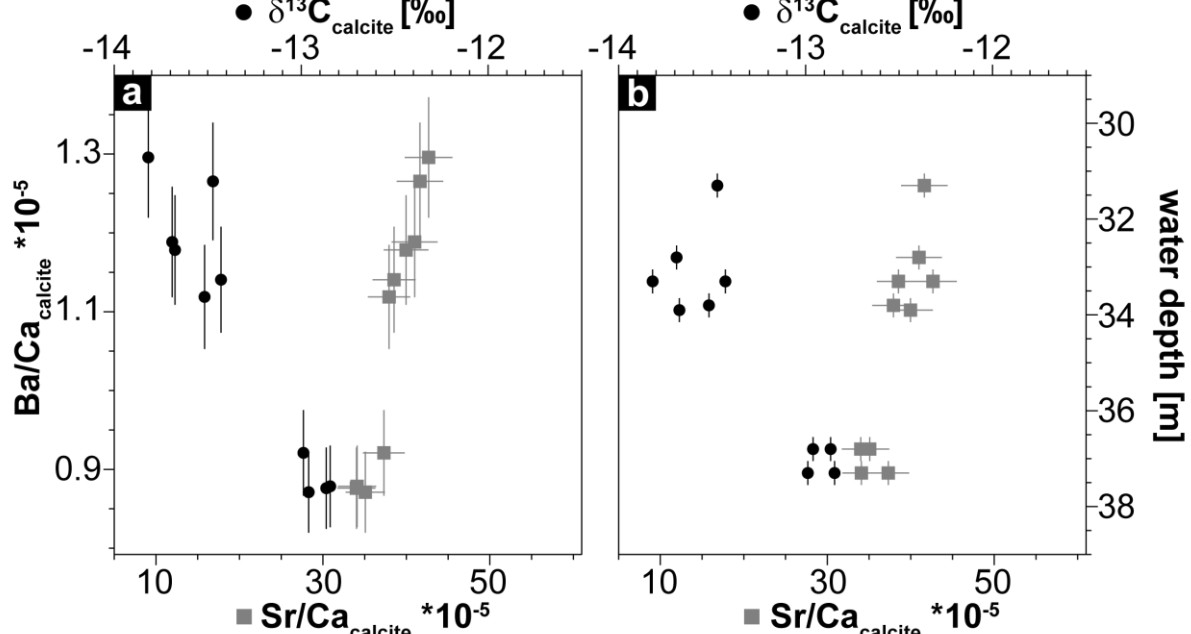

**Fig. 6:** Geochemical data of Hells Bell speleothems showing a strong correlation between Sr/Ca and Ba/Ca ratios ($r^2 = 0.91$) and between Ba/Ca and $\delta^{13}C_{calcite}$ ($r^2 = 0.89$) (a) and a trend of increasing $\delta^{13}C$ and decreasing Sr/Ca with increasing water depth of the samples (b). Given uncertainties represent $2\sigma$ standard deviations and $\pm\ 0.25$ m is assumed as uncertainty for the water depth of the Hells Bell samples.

### 3.4 Turbid layer filtrate

Though the turbid layer appears dense in photographs taken during dives, the water sampled from the turbid layer was clear, with no visible turbidity during sample handling. Though, electron microscopy of the filter reveals that abundant particles were extracted from the turbid layer (Fig. 7 a). Particle sizes range between 1–100 µm, but most are in the range of 1–10 µm. They consist of Ca-carbonate crystals (Figs. 7 c and c1), globular particles consisting of elemental sulfur (Figs. 7 d and d1) and silicate particles of different compositions (Figs. 7 e and e1). Also, numerous intact and broken shells of siliceous diatoms were found on the filter. Some calcite crystals incorporated broken parts of silica shells (Figs. 7 c1).





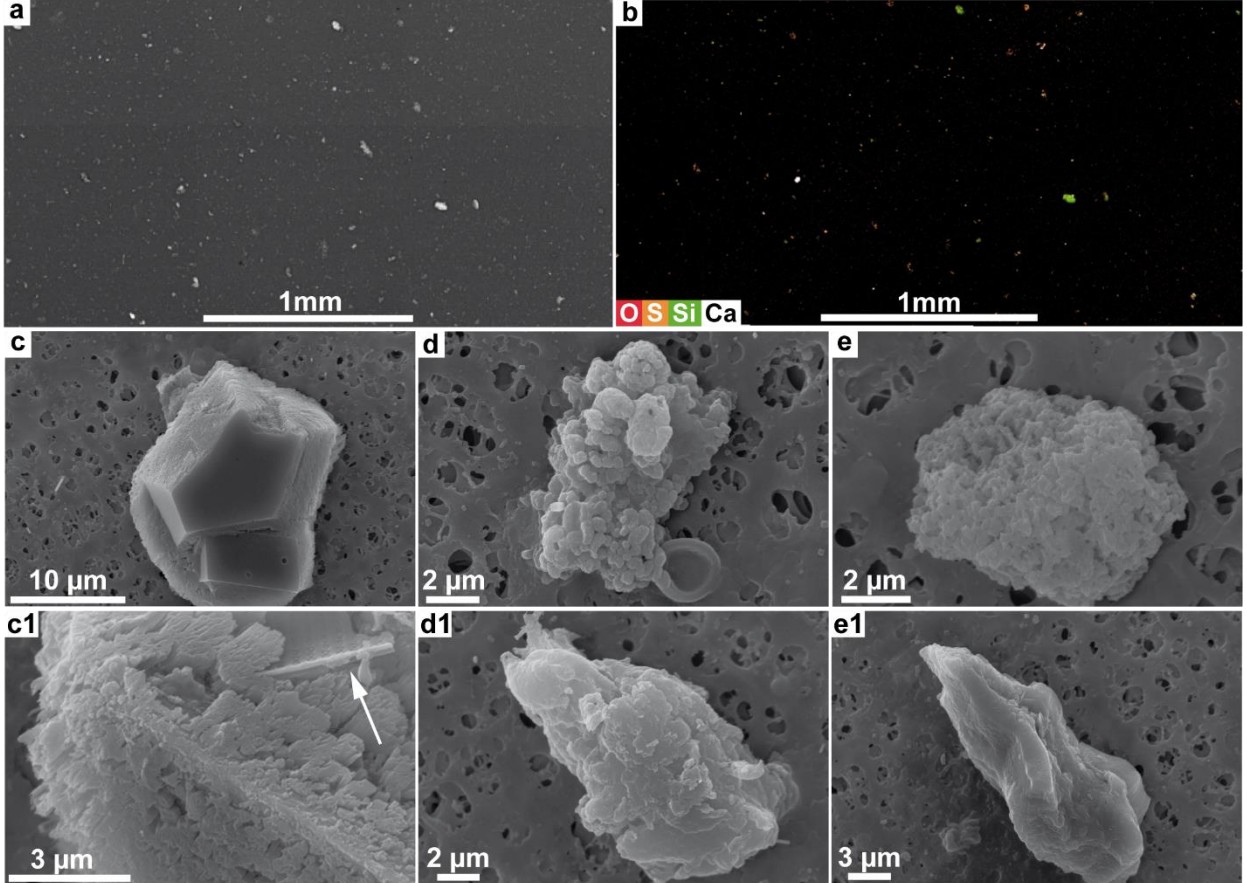

**Fig. 7: SEM-analysis of turbid layer filtrate. Various particles on the filter are visible on the SEM picture of a larger area of the filter (a). An element map for O, S, Si and Ca of the same filter area of (a) is shown in (b) revealing that most particles consist of elemental sulfur, Ca-rich particles and silica particles. EDX analysis of individual particles on the filter verified the particles as Ca-carbonates (c, c1), elemental sulfur (d, d1) and silicate phases (e, e1). The white arrow in c1 points to a fragmentary silica shell.**

## 4. Discussion

The detailed analysis of the water body and the geochemical analyses of Hells Bells presented in this study enables us to infer a comprehensive hypothesis on Hells Bells formation. In our previous study on Hells Bells we already argued that Hells Bell form under water in a lightless environment. We suspected the formation to take place in the lowermost fresh water body above the halocline (Stinnesbeck et al., 2017b). However, due to insufficient hydrogeochemical data we could only speculate on the processes leading to Hells Bells formation.

In the following discussion we deduce that the stagnancy of the water body favors the development of a pelagic redoxcline in which chemolithoautotrophic processes, mainly intense anaerobic sulfide oxidation, creates favorable conditions for calcite precipitation and hence, Hells Bells formation. Furthermore, we discuss the role of a dynamic halocline elevation and infer




the crucial factors for Hells Bells formation in El Zapote Cenote, which may give hints to the apparent exclusivity of these speleothems to only few cenotes in a very restricted area.

## 4.1 Limnological and hydrological conditions in El Zapote Cenote

The water temperature profile (Fig. 3 a) offers valuable clues on the hydrological conditions in the El Zapote cenote. Mixing of the water in the narrow cenote shaft from 0–30 m water depth is indicated by constant temperatures and oxygenation, whereas linearly increasing temperatures in the wide dome-shaped cenote from 30–55 m water depth and linearly decreasing dissolved oxygen concentrations indicate conductive heat transport and oxygen diffusion, respectively (Fig. 3 a). This suggests that the water body from 0–30 m water depth is mixing-dominated and diffusion-dominated from 30–>50 m water depth. This interpretation is also supported by constant EC values in the cenote shaft and constantly increasing EC values from 30 m water depth down to the top of the halocline at 36.8 m water depth (Fig. 3 a). Another indication for stagnant conditions of the water body is the shape of the halocline itself. Compared to other cenotes of the Yucatan Peninsula being deep enough to reach the halocline, El Zapote cenote particularly differs in the extent of the halocline, the transition zone from fresh to salt water. At El Zapote cenote, the halocline is about 10 m thick (Fig. 3a), as compared to a transition zone thickness of 1–5 m of other cenotes of Quintana Roo (Kovacs et al., 2017b); Stoessell et al., 1993).

The constant decrease of DIC, sulfide and orthophosphate below about 40 m water depth indicates a sink of these chemical species into depths greater than the cenote (>54 m water depth). This sink may result from advection of flowing water masses in conduits or zones of intensified hydraulic conductivity in a deeper cave system at around 60 m below the present sea level. Such deep cave systems could have developed during glacial sea level low stands (e.g. Smart et al., 2006).

In general, the water body of El Zapote cenote is stagnant from 30 m water depth down to the bottom of the cave where mass transfer is predominantly due to chemical diffusion. This is essential for the understanding of hydrogeochemistry and the ongoing biogeochemical processes in the El Zapote cenote.

## 4.2 Hydrogeochemical processes in El Zapote cenote

There might be main biogeochemical domains regulating the redox conditions of the meromictic El Zapote cenote, the organic matter-rich sediments of the debris mount and the redoxcline. Therefore, we here discuss the biogeochemical processes occurring in the sediment and in the redoxcline of the water body.

### 4.2.1 Sedimentary biogeochemical processes

The anaerobic conditions and high concentrations of metabolites such as S(-II) and $CH_4$ can be attributed to anaerobic heterotrophic organic matter (OM) decay in the debris mount sediments. Both the debris mount and the cenote floor are covered with a relatively thick layer (~1 m) of OM, mostly leaves and other plant remains, according to the descriptions of the divers. As a consequence of stagnancy in the meromictic water body and oxygen deficiency on the cave bottom, this OM is respired by heterotrophic microorganisms in the sediment via anaerobic fermentative and respiratory pathways.





Anaerobic OM degradation by fermentation and sulfate reducing bacteria produce hydrogen and hydrogen sulfide (S(-II)), $CO_2$ (DIC) and acidity, thus lowering the pH. Elevated concentrations of DIC and S(-II) are found in the halocline (Fig. 3c), and low $\delta^{13}$C-$HCO_3^-$ indicate a microbial origin of the hydrogen carbonate (e.g. Mook, 2000) (Fig. 4). Additionally, pH values are more acidic in the halocline (Fig. 3a and b) and sulfate reduction is further supported by decreasing $SO_4^{2-}$/$Cl^-$

ratios in the halocline of up to 32% compared to seawater ratio of 5.2 (Fig. 3 c) (Stoessell et al., 1993).

Methane-producing archaea (methanogens) metabolize degraded OM releasing $CH_4$ and DIC. Although this pathway is less energy efficient than sulfate reduction, methanogens may dominate in deeper parts of the sediments where sulfate is already consumed. Diffusion of $CH_4$ from the sediment into the water column leads to $CH_4$ concentrations of up to 25 μmol $l^{-1}$ identified in the halocline of El Zapote.

Ammonium is likely released from organic matter degradation in the organic rich sediment and is also released to the water column at the halocline.

Other common anaerobic heterotrophic metabolic pathways in sediments, such as the reduction of iron, are subordinated processes, most likely due to the limitation of iron oxides in the limestone karstic area. The elevated but still exceedingly low amounts of dissolved iron in the halocline as compared to the fresh water body (Fig. S3) are rather not indicative for the

absence of these processes, as iron solubility is limited by the affinity to form iron sulfides in the presence of high amounts of $S^{-II}$.

### 4.2.2 Water column biogeochemical processes

The redoxcline from 35 to 36.8 m water depth coincides with a peak in turbidity which is detectable both visually (Fig. 2 b) and geochemically (Figs. 3 a and b). Dissolved oxygen (DO) concentrations drop to undetectable levels at the top of the

redoxcline, indicating that anaerobic biogeochemical processes prevail within the redoxcline (Fig. 8).

In our previous study we tentatively attributed these conditions to a full heterotrophic redox zonation due to organic matter decomposition (Stinnesbeck et al., 2017b). Fine organic matter accumulates along the density contrast at the top of the halocline and heterotrophic microbial communities thrive from the aerobic and anaerobic decomposition of this organic matter. This is also indicated in the results of this study by minor nitrification from ~34–35 m water depth (Fig. 3 b), non-

linearly decreasing dissolved oxygen contents from ~34–35 m and by slightly more acidic pH values above and in the uppermost part of the turbid layer.

Nevertheless, the more detailed data presented in this study now underline the importance of planktonic chemolithoautotrophic processes in the pelagic redoxcline which are driven by the upward diffusion of reduced sulfur, carbon and nitrogen species released from the anaerobic degradation of organic material at the cenote floor. Pelagic

redoxclines develop in density stratified marine (e.g Berg et al., 2015) as well as lake environments (e.g. Noguerola et al., 2015). In redoxclines below the photic zone the microbial community is dominated by chemolithoautotrophs, with a considerable amount of chemoautotrophic production and dark carbon fixation (e.g. Grote et al., 2008; Jørgensen et al.,



1991; Jost et al., 2010; Noguerola et al., 2015). The development of pelagic redoxclines was also reported for deep density stratified cenotes of the Yucatán Peninsula (e.g. Socki et al., 2002; Stoessell et al., 1993; Torres-Talamente et al., 2011) .

In our previous study members β-*proteobacteria Hydrogeophilaceae* and the ε-*proteobacteria* genus *Sulfurovum* were reported as dominant within the aqueous microbial community. Most members of these bacterial groups are chemolithotrophic or mixotrophic using reduced sulfur compounds or hydrogen as electron acceptors and oxygen or nitrogen-compounds as electron acceptors (Stinnesbeck et al., 2017b).

The white cloudy turbid layer could be the result of a dense accumulation of these microorganisms e.g. sulfur-oxidizing bacteria, analogous to that reported for Bundera sinkhole in Australia (Seymour et al., 2007). Elemental sulfur particles or polysulfides were detected on the turbid layer filtrate and indicate sulfur oxidation in the turbid layer or redoxcline (sec 3.4 and Fig. 7); these particles are formed as intermediates in the microbial oxidation of sulfide (Findlay, 2016).

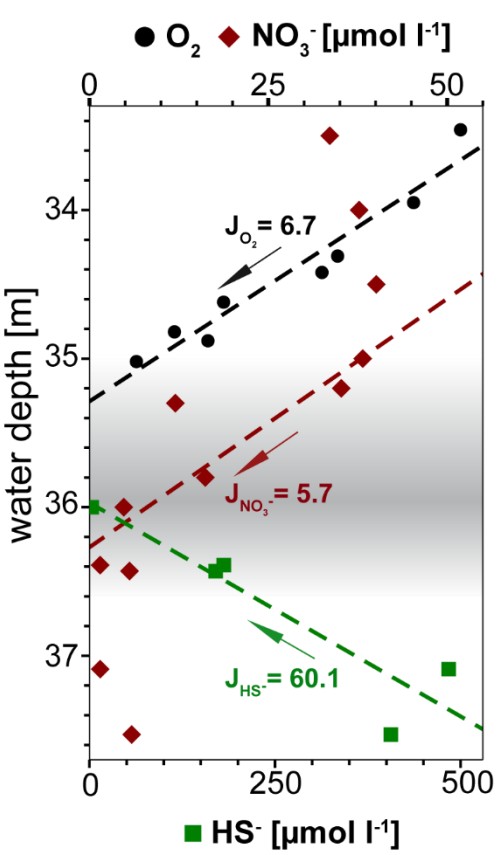

**Fig. 8: Concentration profiles of dissolved $O_2$, $NO_3^-$ and $HS^-$ (calculated with PhreeqC) in water depths around the redoxcline. The fluxes J are given in $10^{-8}$ µmol m$^{-2}$ s$^{-1}$. The linear fit of $O_2$ and $HS^-$ is calculated for the range of plotted values, while for $NO_3^-$ it is calculated only for the range from 34.4–36.6 m water depth. Only $O_2$ values above detection limit (6.3 µmol l$^{-1}$) from winch profile 2 were considered for the calculation (Table S1).**

The oxidation of sulfide in the redoxcline is likely anaerobic, as sulfide vanishes at around 36 m while dissolved oxygen is already at undetectable levels at 35 m water depth and both concentration profiles are not overlapping (Fig. 8). Furthermore,




the oxygen flux towards the redoxcline is around one magnitude lower than the flux of the reduced sulfur species HS$^-$ indicating that sulfide oxidation via aerobic pathways is minor (Fig. 8). Thus, sulfide oxidation within the redoxcline must be predominantly via anaerobic pathways. As the downward flux of NO$_3^-$ towards the redoxcline intersects with the upward flux of HS$^-$ (Fig. 8), assimilatory anaerobic sulfide oxidation could be obtained with nitrate as terminal electron acceptor

producing elemental sulfur and nitrogen under the consumption of protons. The overall mass-balanced energy generating reaction for chemoautotrophic nitrate-driven anaerobic sulfide oxidation (ND-SO) is given in reaction (R1):

$$7HS^- + 2NO_3^- + CO_2 + 9H^+ \rightarrow 7S^0 + N_2 + CH_2O + 7H_2O \tag{R1}$$

According to reaction (R1) ND-SO could account to one third of the HS$^-$ oxidation, despite the flux of NO$_3^-$ towards the redoxcline is around on order of magnitude lower than the HS$^-$ flux (Fig. 8). Furthermore ND-SO is acid consuming and

sulfide oxidation to elemental sulfur is more acid consuming than the full sulfide oxidation to sulfate (see also Visscher and Stolz, 2005). The abundance of elemental sulfur particles found in the turbid layer filtrate (Fig. 7) indicates that sulfide oxidation to elemental sulfur is predominant. Full oxidation of sulfide to sulfate is less likely as no increase of sulfate is observed in the redoxcline (Fig. 3 c). Maxima in pH are known to occur when sulfide is oxidized to elemental sulfur with nitrate as electron acceptor (Kamp et al. 2006). In consequence, the minimum of nitrate in the redoxcline and slight alkaline

pH shift, indicate that ND-SO is a relevant process in the redoxcline (Fig. 3 b and c, Fig. 8). Therefore, the proton consuming ND-SO could be the biogeochemical process in the redoxcline creating a disequilibrium in the carbonate dissolution-precipitation reaction, favoring calcite precipitation. This mechanism was recently reported for the formation of stromatolites below the photic zone of the Arabian Sea. There, a collective effect of proton-consuming ND-SO and alkalinity-producing sulfate driven-oxidation of CH$_4$ (SD-OM) leads to authigenic carbonate precipitation in microbial mats in the vicinity of

CH$_4$-seeps (Himmler et al., 2018).

Anaerobic SD-OM (e.g. Bailey et al. 2009) is likely to occur at the redoxcline, as dissolved CH$_4$ concentrations vanish at around the same depth of sulfide (~36.5 m) and $\delta^{13}$C-CH$_4$ values show a strong peak towards higher values at the same water depth (Fig. 4).

Autotrophy also supports calcite precipitation by taking up CO$_2$ for the synthesis of biomass (Castanier et al., 1999; Kosamu

and Obst, 2009). Although a decrease of DIC is not observed at the redoxcline, chemolithoautotrophy is indicated by the $\delta^{13}$C-HCO$_3^-$ in the water body (Fig. 4). The peak of higher values in the redoxcline indicates inorganic carbon assimilation by microorganisms (dark-CO$_2$ uptake). As organisms usually prefer to metabolize $^{12}$C (it takes less energy to break the $^{12}$C bond instead of $^{13}$C) they effectively consume HCO$_3^-$ with lower $\delta^{13}$C values, which subsequently results in higher $\delta^{13}$C-HCO$_3^-$ values in the remaining dissolved inorganic carbon. Hence, the peak towards more positive $\delta^{13}$C-HCO$_3^-$ values

identified in the redoxcline of El Zapote at ~36 m water depth, may be attributed to microbial CO$_2$ assimilation or dark CO$_2$ fixation.



### 4.3 Hypothesis on Hells Bells formation

It was shown before that Hells Bells form within the freshwater indicated by $\delta^{234}U_{initial}$ values of 16–25 ‰ of the Hells Bells calcite (Stinnesbeck et al., 2017b). The depth zone of Hells Bells formation within the fresh water layer can now be narrowed down by the application of given distribution coefficients D(Mg) of the temperature-dependent partitioning of Mg

into calcite in Eq. (2):

$$(Mg/Ca)_{solution} = \frac{(Mg/Ca)_{solid}}{D(Mg)} \qquad (2)$$

Applying the mean value of Mg/Ca$_{solid}$ determined for Hells Bells calcites (Table 1) and D(Mg) at 25°C given by Huang and Fairchild (2001) and Rimstidt et al. (1998), the calculation of Mg/Ca of the solution from which the Hells Bells calcite precipitated yields a Mg/Ca$_{solution}$ of 0.73 and 1.06, respectively. Mg/Ca$_{solution}$ ratios in this range are found in the water of the

redoxcline and the uppermost top of the halocline in 36–37 m water depth, thus supporting the interpretation that Hells Bells formation takes place in the redoxcline (Fig. 3 c and Table S4).

The calcite crystals found in the turbid layer filtrate give further hints on calcite precipitation in the redoxcline (Fig. 7 c). It is not yet known whether these particles represent autochthonous matter of the turbid layer. Nevertheless, formation of calcite crystals at the density boundary is likely, as fine particulate matter is accumulated there and may act as crystallization seeds.

This process is indicated by calcite crystal formation around silica shells (Fig. 7 c1). The high sulfur contents found in Hells Bells calcite also supports this assumption as small sulfur particles are abundant in this water layer and are easily enclosed in calcite crystals growing there (Table 1).

Based on the indications of Hells Bells formation in the redoxcline and taking the biogeochemical processes discussed before into account we propose the following scenario illustrated in Figure 9. It summarizes the biogeochemical processes

inducing calcite oversaturation and calcite precipitation in the turbid layer and the redoxcline of El Zapote cenote. Heterotrophic bacterial decomposition of organic matter in sediment of the debris mount releases $CO_2$ ($HCO_3^-$), nutrients ($P_{ortho}$) and reduced species of sulfur (S(-II)) and nitrogen ($NH_4^+$). Due to the stagnant conditions in the cenote, these species are transported via diffusion, thereby allowing for the formation of a defined and stable redoxcline. Here, anaerobic chemolithoautotrophy, and especially proton consuming nitrate-driven sulfide oxidation, increase alkalinity, thus favoring

calcite precipitation (Fig. 9). The required $Ca^{2+}$ ions for calcite precipitation are constantly supplied to the redoxcline by upward diffusion from the Ca-enriched saline water body (Stinnesbeck et al., 2017b).





## Biogeochemical processes involved in Hells Bells formation

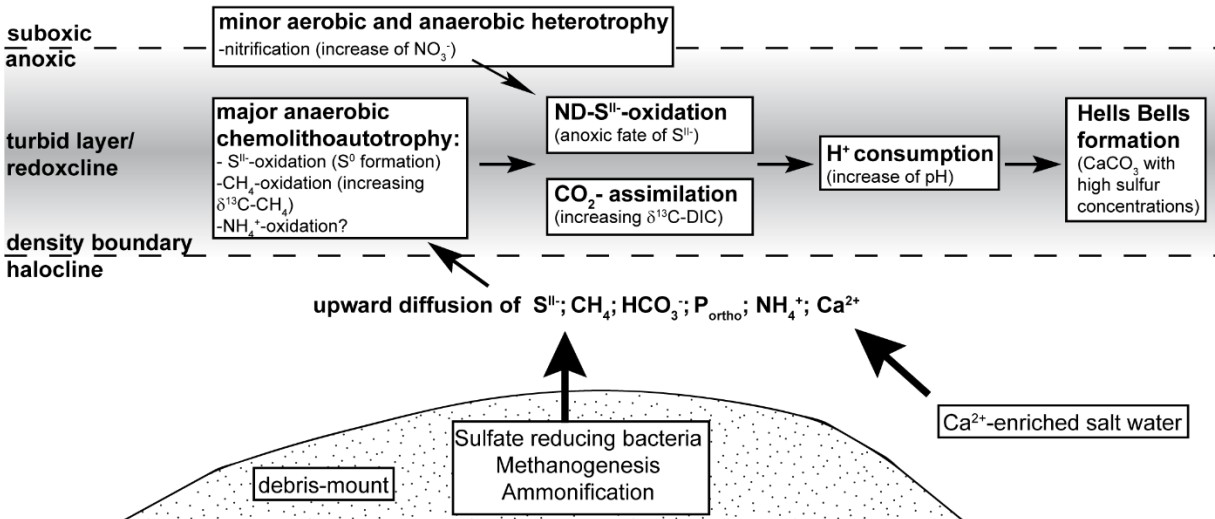

**Fig. 9: Scheme of the biogeochemical processes involved in the sediment and redoxcline of the water column of the El Zapote cenote that lead to Hells Bells formation.**

However, biogeochemical processes in the redoxcline alone are not sufficient to explain the shape of Hells Bells nor their occurrence over a wide vertical zone of around 10 m water depth. This implies that another essential process must be involved in Hells Bells formation, which is the repeated elevation of the halocline.

### 4.3.1 The role of halocline elevation in Hells Bells formation

Hells Bells formed in modern to historic times and occur in a relatively wide vertical zone of about 10 m from 28–38 m water depth (Stinnesbeck et al., 2017b). This indicates that their underwater growth occurred under environmental conditions similar to the ones detected by us, as modern sea water levels were already reached at about 4.5 ka BP (Hengstum et al., 2010) and thus significantly earlier. Modern Hells Bells therefore precipitate either permanently in the entire depth zone reaching from 28-38 m, or in the narrow 1–2 m wide redoxcline or turbid layer above the halocline (Fig. 9). According to the data presented here the latter hypothesis appears much more likely to us. Therefore, we propose that growth of Hells Bells is a non-permanent episodic process which majorly depends on a highly variable halocline elevation in the cenote (Fig 10). The halocline position is a function of hydrostatic pressure of the overlying fresh water layer, with a general trend of increasing depth with increasing distance to the coast (Fig 10 a) (e.g. Bauer-Gottwein et al., 2011). Extraordinary recharge events (e.g. hurricanes) must have a significant effect on the depth position of this layer (Fig. 10. b). During these events of enormous precipitation, the halocline is temporarily pushed downwards by the amount of new the fresh water infiltrating into the Yucatán karst aquifer. Escolero et al. (2007) detected a halocline rise of 17.5 m in a Yucatán aquifer well and ascribed this to a piston-like effect. During recharge events, the water flow is predominantly vertical and salt water is temporarily pushed down. In the period after these events, the halocline bounces back and oscillates until a static equilibrium is again reached.





Regional and local vertical and lateral hydraulic transmissivities of both the epi- and the phreatic karst can also result in spatially variable hydraulic pressure of the fresh water lens (Williams, 1983), thus leading to a lowered halocline beneath areas of higher, and an elevated halocline beneath areas of less vertical hydraulic transmissivity. El Zapote cenote is located in the Holbox fracture zone that in the area is characterized by N-S trending lineaments of increased permeability (Bauer-Gottwein et al., 2011, and references therein). Hurricanes that pass the area frequently (Farfán et al., 2014) will therefore lead to episodic elevation of the halocline (Fig. 10).

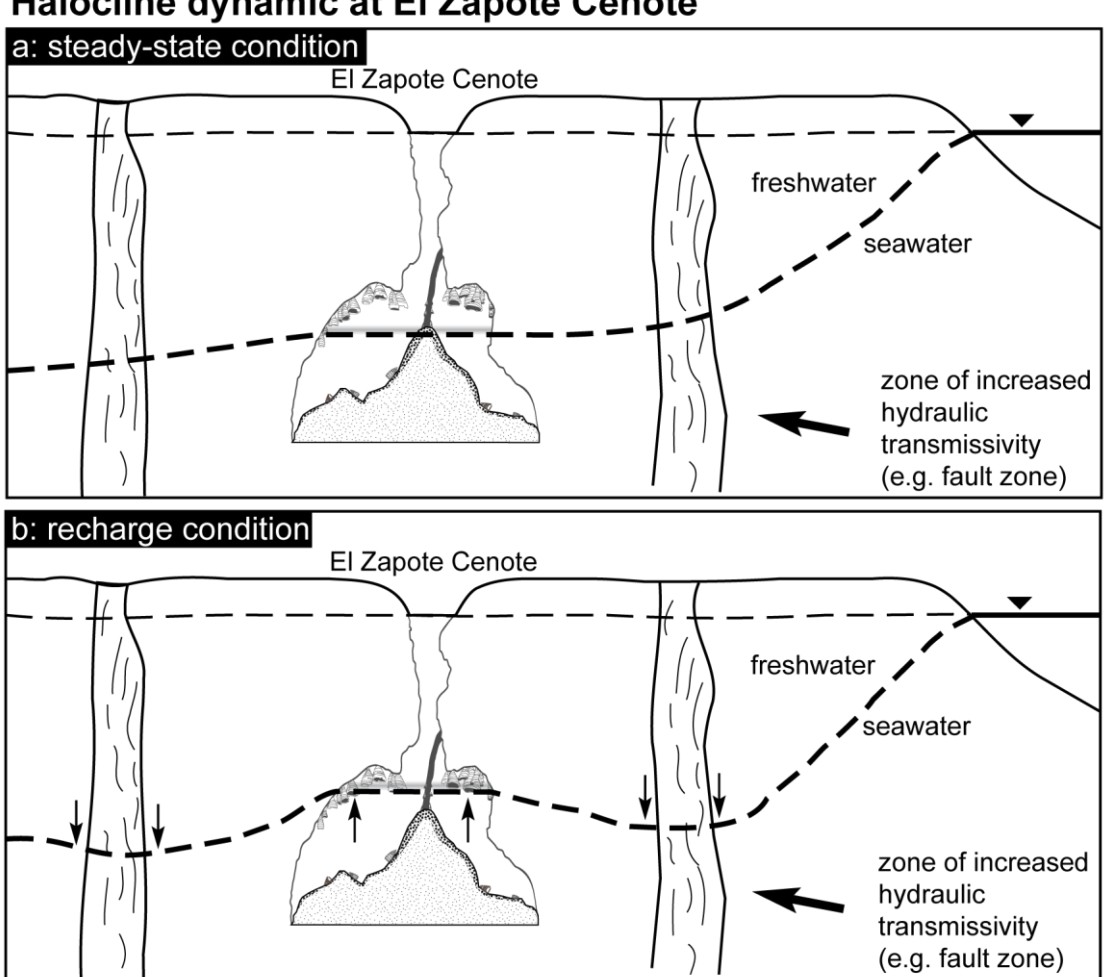

**Fig. 10: Sketch of dynamic halocline elevation within the Yucatán Karst aquifer. Halocline depth increases with increasing distance from the coast in a steady-state condition (a). Recharge events (e.g. hurricanes) result in a lower halocline beneath areas of high vertical transmissivities and an elevated halocline in areas of low hydraulic transmissivity, e.g. El Zapote cenote (b).**

Although there is yet a lack of data to allow for a positive identification of El Zapote as a recharge-driven cenote, the process explains the presence of Hells Bells in a zone of 28–38 m water depth. During an episodic rise of the halocline, the zone of Hells Bells formation (redoxcline) also rises, thus inducing calcite formation in a comparably shallow water depth (Fig. 10).





Our model also explains the alternating presence of layers of dog-tooth calcite and microcrystalline calcite indicating dissolution (Stinnesbeck et al., 2017b). During episodes of an elevated halocline, dissolution of Hells Bells may occur in the lower depth range due to the intermittent increase of sulfide-rich and carbonate-undersaturated water.

A dynamic halocline depth position at El Zapote cenote is also supported by the positive correlation with water depth of
Sr/Ca and a negative correlation of $\delta^{13}$C of the Hells Bells calcite (Fig. 6). Hells Bells formed in lower water depths show slightly higher contents of the trace elements Sr, Mg and Ba and slightly lower $\delta^{13}$C values than Hells Bells formed in greater water depths (Table 1 and Fig. 6). The higher incorporation of the trace elements Sr, Mg and Ba is either obtained by faster growth rates (Tesoriero and Pankow, 1996), or by elevated concentrations in the solution from which the calcite is precipitated. The latter process is more likely, as the amount of seawater increases in the turbid layer when the halocline is
located at lower water depths. Lower $\delta^{13}$C values support this assumption, as lowest $\delta^{13}$C-$HCO_3^-$ values are detected in the halocline of the modern El Zapote cenote (Fig. 4).

The increase in seawater in the lowermost fresh water and the turbid layer could result from turbulence induced by halocline elevation as a reaction to recharge events (Kovacs et al., 2017b). Minor mixing of the water bodies would be sufficient to increase the concentrations of Sr, Mg, Ba and decrease $\delta^{13}$C-$HCO_3^-$ in the turbid layer, as seen at El Zapote.

Following this model of a hydraulically-driven halocline elevation, the question remains why Hells Bells are restricted to a zone of 28–38 m water depth. This range could solely depend on the hydraulic conditions, e.g. Hells Bells formation reflecting maximum and minimum elevations of the halocline during recharge events. The lower (38 m) level of Hells Bells formation may represent the stable environmental conditions in the modern El Zapote cave, influenced only by the thickness of the freshwater body and the mean sea level. The upper range boundary, on the other hand, could well be given by the
shape of the sinkhole and limnological conditions in the narrow cenote shaft reaching from 0–30 m water depth. In this latter unit the water body is mixing- rather than diffusion-dominated (Fig. 3. Section 4.1). A rise of the halocline to about 28 m water depth would therefore lead to an exposure to fast and convective oxygen supply from the mixed-in fresh water body above, and consequently to aerobic microbial sulfide oxidation, which is an acid-producing reaction (e.g. Jones et al., 2015). Hells Bells formation would then stop as it is tied to the anaerobic ND-SO. The occurrence of a zone of brown-colored
manganese oxide coatings on Hells Bells and cave wall at and above 30 m water depth, indicates that the redoxcline must temporarily have reached up to this level (Fig. 2 c, d and Fig. S4). Manganese dissolved in the halocline and turbid layer was then oxidized to manganese oxide precipitates (Fig. S4).

The dynamic history of halocline elevation at El Zapote cenote cannot be resolved to date but raises an interesting issue of further research on the dynamic hydraulic response of the Yucatán aquifer to extraordinary recharge events, especially as this
process could be a key factor for the formation of Hells Bells. We are currently addressing this issue by logging the hydraulic head of the fresh water body and the electrical conductivity at a fixed position in the halocline of El Zapote cenote.





### 4.3.2 Shape of Hells Bells

Previously, we attributed the growth of Hells Bells to microbial mediation (Stinnesbeck et al., 2017b). We hypothesized that autotrophy (ammonia oxidation) and denitrification are the main factors that trigger calcite precipitation at the surface of the Hells Bells and that calcite precipitation could further be supported by the presence of negatively charged extracellular

polymeric substances (EPS), leading to the accumulation of $Ca^{2+}$ ions and to supersaturation of calcite within biofilms (e.g. Dupraz et al., 2009). However, the large size and form of the dog-tooth calcite crystals of Hells Bells rather resemble slow growing inorganic calcite crystals rather than biologically-mediated precipitates (Fig.2 e and Fig. 5). This hypothesis is supported by Bosak & Newman (2005) who investigated microbial kinetic controls on calcite morphology and found that microbially mediated calcite precipitated at low calcite supersaturation shows more anhedral crystal morphologies, compared

to the more euhedral abiotic ones. Although the microbial activity in the redoxcline induces calcite oversaturation, the hypothesis presented in this study is compatible with an inorganic calcite precipitation of the Hells Bells from a biologically mediated water layer.

Ultimately, the hypotheses of dynamic halocline elevation and biogeochemically induced calcite precipitation in the redoxcline can be integrated. Hells Bells grow downward and are conically divergent, with a strict horizontally lower margin

and a hollow interior. Specimens also tend to be oriented towards the cenote center (Stinnesbeck et al., 2017b). The horizontal downward growth is indicative for a precipitation from a defined layer within the water column (i.e. the redoxcline). Also, an abrupt elevation of the redoxcline as a response to recharge events and a subsequent decelerated drop towards its original position serves to explain the downward growth of Hells Bells. This is indicated by the tendency towards downward orientation of the calcite crystal growth axis (Fig. 2 e). The fact that Hells Bells specimens growing on the

inclined cave wall are always oriented towards the cenote center could result from a lateral gradient in the chemolithoautotrophic intensity. The "energy source" (i.e. sulfide) used by the chemolithotrophic microbial community in the redoxcline is the release of reduced carbon, sulfur and nitrogen species from anaerobic organic matter decay in the organic-rich sediments on the debris mount. Both the morphology of the cenote and the diffusive mass transport likely result in radial concentration gradients of upwards diffusing reduced species from sediment of the debris mount. These conditions

limit the availability of reduced species in locations of the redoxcline distal to the debris mount, and vice versa. Consequently, the intensity of chemolithoautotrophy and hence calcite oversaturation is preferentially higher in the center proximal to the debris mount and decreases towards the cenote walls. This accounts for both the inclined bells as well as for horse-shoe like horizontal openings of Hells Bells which always face towards the wall.

Nevertheless, we can also not exclude the potential influence of microorganisms forming a biofilm community on the

surface of Hells Bells. Stinnesbeck et al. (2017b) showed that this community does not resemble the planktonically growing microbial biocenosis but forms a distinct community that seems to thrive catalyzing the reduction and oxidation of different nitrogen species. However, it is not known to date to what percentage the activity of these organisms contributes to the shape of the speleothems.



## 4.4 Prerequisites for the formation of Hells Bells

Hells Bells have so far been identified in a few cenotes only of a restricted area of the North-Eastern Yucatán Peninsula (Stinnesbeck et al., 2017b), although the peninsula hosts many thousands of sinkholes (Bauer-Gottwein et al., 2011). The question thus arises which factors are needed for the generation of these underwater speleothems. The following apparent

prerequisites for Hells Bells formation appear likely to us:

- The cenote or sinkhole must be deep enough to reach the halocline in order to have a density stratified water column (meromixis).
- Sufficient input of organic material to the cenote bottom is required to create anoxia in the halocline with a release of reduced sulfur, carbon and nitrogen species.
- A meromictic stagnant water body is needed that allows for the formation of a redoxcline in which anaerobic chemolithoautotrophy prevails in a lightless environment. This leads to a narrow zone of calcite oversaturation in the water body.
- Special hydraulic conditions are needed which allow the halocline to rise and fall in order to form subaqueous speleothems.

## 5. Conclusion

The unique underwater speleothems termed Hells Bells recently described from El Zapote west of Puerto Morelos on the northern Yucatan Peninsula, Mexico, are most likely formed in the redoxcline, a narrow layer in the lowermost fresh water body immediately overlying the halocline. We propose a biogeochemical mechanism for the formation of these structures, that induces calcite oversaturation favoring calcite precipitation within the redoxcline. The upward diffusion of reduced

sulfur, carbon and nitrogen stimulates a chemolithoautotrophic microbial community thriving above the halocline at El Zapote cenote. Chemolithoautotrophy and proton-consuming nitrate-driven anaerobic sulfide oxidation lead to calcite precipitation, and hence Hells Bells formation, in a narrow depth zone confined to the redoxcline, or turbid layer. We further postulate a dynamic elevation of the halocline as an episodic hydraulic response to recharge events, e.g. hurricanes, that accounts for Hells Bells occurrence over a vertical range of 10 m water depth.

**Video supplement**

https://doi.org/10.5446/39353

**Team list**

Simon M. Ritter – SR

Margot Isenbeck-Schröter – MIS

Christian Scholz – CS

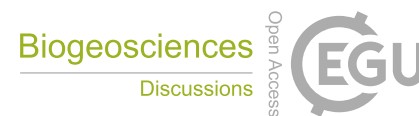

Frank Keppler – FK

Johannes Gescher – JG

Lukas Klose – LK

Nils Schorndorf – NS

Jerónimo Avilés Olguín – JAO

Arturo González-Gonzalez – AGG

Wolfgang Stinnesbeck – WS

Dirk Penzel – DP

Christine Loew – CL

Eugenio Aceves Núñez – EAN

Thomas Vogt – TV

Alexander Varychev – AV

Gregor Austermann – GA

Anne Hildenbrand – AH

**Author contribution**

WS and AGG initialized the project and funding was acquired by WS, AGG and MIS. Sampling was planned and conducted by SR and CS. Underwater sampling was conducted by JAO, DP, CL and EAN, while underwater videographer TV was documenting part of the sampling producing the footage for the video supplement. Instrumentation and methodology was provided by MIS for hydrogeochemical analyses, FK for gas and stable carbon isotope analytics, AH and GA for light

microscope-imaging and AV for SEM-imaging and GA and. SR, LK and NS collected the data and CS, MIS, FK and AV validated it. SR, CS, MIS, LK, NS interpreted the results. SR developed the hypothesises equally supported by CS and MIS. SR visualized the data and prepared the original draft, MIS supervised. JG, WS and FK critically reviewed the manuscript.

**Competing interests**

The authors declare that they have no conflict of interest.



**Disclaimer**

**Acknowledgements**

We gratefully acknowledge the owners of Cenote Zapote-Ecopark Mrs. Rosario Fátima González Alcocer and Mr. Santos Zuñiga Roque and their Ecopark team members Daniel de Jesus Tum Canul, Israel Mendez Castro and Eunice Mendez
Castro de la Cruz for granting us access to the cenote and their great support during field work. We would like to thank the technical cave divers Eugenio Aceves Núñez, Christine Loew, Dirk Penzel and Thomas Vogt for their excellent work in retrieving samples from El Zapote cenote. Many thanks to Vicente Fito, the original Zapote-cave explorer for sharing his discovery. We thank Markus Greule, Bernd Knape, Stefan and Silvia Rheinberger and Swaantje Brzelinski for conducting geochemical analyses and many discussions that helped to produce and improve this dataset. We are grateful for the help of
Alexander Varychev, Anne Hildenbrand and Gregor Austermann with the optical analyses and especially Master student Tianxiao Sun for examining thin sections. We are grateful for the effort of the technicians Christian Mächtel and Andreas Thum of the Institute of Earth Sciences for their technical and constructional support prior to the field trip. Finally, we thank Jan Hartmann, Andrea-Schröder-Ritzrau, Tobias Anhäuser and Daniela Polag for their sincere proof-reading and many fruitful discussions.
This work was funded by CONACYT-FONCICYT-DADC / 000000000278227 and the Deutsche Forschungsgemeinschaft DFG (STI128/28 and STI128/36).

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





**Table 1: Geochemistry of samples from the lowermost tips of Hells Bells growing on a subfossil ceiba tree that fell into the El Zapote cenote about 3500 cal yr BP. Individual tree bells were sampled at water depths from 31.3 to 37.3 m. The lack of samples in water depths from 34.3 to 36.8 m water depth is due to poor visibility in the turbid layer above the halocline (compare Fig. 1b). Given uncertainties represent 2σ standard deviations.**

| Sample No. | water depth [m] $^{\alpha}$ | Mg/Ca [*10$^{-3}$] | Sr/Ca [*10$^{-5}$] | Ba/Ca [*10$^{-5}$] | Fe/Ca [*10$^{-5}$] | Mn/Ca [*10$^{-6}$] | S/Ca [*10$^{-3}$] | δ$^{13}$C [‰$_{VPDB}$] |
|---|---|---|---|---|---|---|---|---|
| 1 | 31.3 | | | | | | | |
| 2 | 31.3 | 21.4 | 41.6 | 1.27 | 4.3 | 21 | 2.78 | -13.47 ± 0.01 |
| 3 | 32.8 | 25.6 | 41.0 | 1.19 | 5.2 | 26 | 2.69 | -13.69 ± 0.01 |
| 4 | 32.8 | | | | | | | |
| 5 | 33.3 | 22.0 | 42.7 | 1.30 | 3.8 | 24 | 3.05 | -13.82 ± 0.01 |
| 6 | 33.3 | 22.3 | 38.5 | 1.14 | 3.0 | 16 | 2.54 | -13.43 ± 0.01 |
| 7 | 33.8 | | | | | | | |
| 8 | 33.8 | 20.8 | 37.9 | 1.12 | 3.5 | 24 | 2.72 | -13.52 ± 0.01 |
| 9 | 33.9 | 23.2 | 40.0 | 1.18 | 3.9 | 22 | 2.76 | -13.68 ± 0.01 |
| 10 | 33.9 | | | | | | | |
| 11 | 36.8 | 22.3 | 34.0 | 0.88 | 11.3 | 39 | 3.14 | -12.87 ± 0.02 |
| 12 | 36.8 | 21.1 | 34.1 | 0.88 | 10.6 | 28 | 3.02 | -12.85 ± 0.00 |
| 13 | 37.3 | 24.0 | 37.3 | 0.92 | 6.2 | 32 | 3.17 | -12.99 ± 0.01 |
| 14 | 37.3 | | | | | | | |
| | Mean | 22.5 | 38.6 | 1.10 | 5.76 | 25.8 | 2.87 | -13.37 |
| | 2σ | 2.9 | 5.9 | 0.31 | 5.84 | 12.8 | 0.42 | 0.70 |