# Peer review of "Subaqueous speleothems (Hells Bells) formed by the interplay of pelagic redoxcline biogeochemistry and specific hydraulic conditions in the El Zapote sinkhole, Yucatán Peninsula, Mexico"

_Biogeosciences, 2018_

## Referee Comment (RC1) · Anonymous Referee #1 · 18 Feb 2019

Having read the earlier work of this group on the Hells Bells features, I was delighted to read this manuscript. I general, I think the authors have put together an interesting study on some very curious features (the Hells Bells) and I look forward to seeing this study published in due course with minor revisions. I have only two major comments, with the rest being covered in the line-by-line comments, which I've copied below. First, I think the authors could expand the introduction beyond speleothems to draw in a wider readership. Hells Bells have been widely speculated about informally

by karst researchers for several years, but I think that there is broader biogeochemical story that would be of interest to critical zone and coastal aquifer scientists. The authors don't have to take my suggestion in order for me to recommend publication, but I do think it would help broaden the appeal of the work. The second comment is a bit more substantive. I was not convinced that Hurricane activity would be likely to be responsible for sustained upward migration of the mixing zone necessary for Hells Bells to precipitate in areas above the modern redoxcline. The authors might explore droughts, which are well-documented in the Holocene (see Hodell et al., 2001, reference included in comments below), as an alternative process. Droughts would thin the freshwater lens and elevate the mixing zone for prolonged periods of time needed for the slow precipitation of calcite mentioned in the paper to form large Hells Bells features.

Abstract: Line 20: Given that this is a sinkhole, I'm uncomfortable with the author's description of this chamber as lightless in the absence of measurements of light intensity over annual timescales. While it may appear dark to divers, there may be some "light" that still makes it to the chamber during some portions of the year. Line 32: It would be nice if the authors could include information about why Hell's Bells are not more widely distributed in coastal aquifers in general (and the Yucatan more specifically) in the abstract. Improved information and word economy in the preceding sections should create space for this insertion. Page 2 Line 1: I've been working on karst research problems for almost 20 years and have never heard the term "pending speleothems." That doesn't mean the term does not exist, but it probably means it is not common. Page 3 lines 18-19: I don't understand why a recharge event would elevate the freshwater lens. It should depress it, as recharge would increase the thickness of the lens. . . .. The manuscript that was cited wasn't much help in explaining the process either. Page 10, Line 3: I suggest "oxygenated" or "aerobic" as an alternative to "oxygenized" Page 11, Figure 3: The symbols change in panel C but there is no corresponding legend. I'm guessing that the circles are still winch collected and the squares are diver collected, but it would be helpful if the symbols remained constant across each panel. Page 17

line 9: In this context, "though" is being used to contrast ideas and should not begin a sentence. I suggest restructuring this sentence and the previous one. Page 18 line 9: Bells should be plural. Page 18 lines 13-15: I can agree that sulfide oxidation drives calcite precipitation but I'm still clear on why these features are only found here and not in any of the other sulfidic halocline caves in the Yucatan (or elsewhere), but hopefully this is discussed later in the text. . ... Page 19 lines 23-24: I find these sentences super confusing. Can you please rephrase them? Page 19 line 28: change to "debris mound" Page 20 lines 6-8: I think the authors might want to make this section more clear by saying that methanogenesis in the presence of sulfate isn't expected, but that the methanogenesis could be occurring in the sediments where sulfate is reasonably expected to be depleted. I would then recommend they cite a manuscript or two from the marine sediment pore water research community to demonstrate this depletion is common. The authors' interpretation is also supported by the concentration profile, a point which should be emphasized in the discussion. Page 20, line 13: It might be more clear to say that Fe is only present in low concentrations in limestone and there are limited/no sources of siliciclastic materials that could contribute iron in this part of the Yucatan. Fe-oxides are common in many karst areas where the Fe is derived from weathering of siliciclastics. Page 24, Section 4.3.1 (entire section) – I find the role of Hurricanes in elevating the halocline to be a highly implausible explanation for the vertical zonation of the Hells Bells features. Hurricanes are infrequent and the "piston induced oscillation" effect described by the authors would be short lived. I find it hard to believe that any potential upward displacement by this mechanism would be of sufficient duration to allow for the precipitation of large masses of carbonate photographed above the modern redoxcline. I suggest that a more plausible scenario is that periods of drought (which are well-documented over the Holocence) lowered the water table and increased the elevation of the redoxcline for sustained periods of time (see Hodell, D.A., Brenner, M., Curtis, J.H. and Guilderson, T., 2001. Solar forcing of drought frequency in the Maya lowlands. Science, 292(5520), pp.1367-1370.) Page 27, line 19-20: I still wonder if the general orientation of all of the speleothems towards

the center of the cenote indicates some phototaxic behavior. It would certainly be the simplest explanation and I would encourage the authors to measure light intensity over an annual cycle to see if perhaps small amounts of light are penetrating (not having visited this particular site, I'm unsure how likely this is, but the Cenote profile included in Fig 3, coupled with the fact that the cenote is sufficiently large in diameter to permit cavern tours, suggests that the entrance may be sufficiently large to allow some small amounts of light during some parts of the day/year). Page 28: Section 4.4 prerequisites for formation of Hells Bells. Many cenotes in the Yucatan meet these criteria yet lack Hells Bells. I wonder if one of the more unique aspects of El Zapote is the relatively thick mixing zone, which the authors highlight in their discussion. Many of the haloclines are much thinner, typically due to high flux of fresh or saline water (though I was surprised by how thin the halocline at Cenote Angelita was in the supporting literature – I wouldn't have guessed there was much flow in that location). I suppose many of the haloclines with high fluxes of organic material also have much greater light penetration (such as El Pit and Cenote Angelita), so the location of the mixing zone in a dark cenote (I still hesitate to say that this is completely dark) is also likely important.

---

## Author Comment (AC1) · 25 Feb 2019

First, the authors are very grateful for the constructive comments and suggestions by anonymous referee #1 and fully appreciate the time and effort taken for reviewing the manuscript. We addressed all issues raised by the referee in a point by point response below.

Referee's major comments:

Referee: First, I think the authors could expand the introduction beyond speleothems to draw in a wider readership. Hells Bells have been widely speculated about informally by karst researchers for several years, but I think that there is broader biogeochemical story that would be of interest to critical zone and coastal aquifer scientists. The authors don't have to take my suggestion in order for me to recommend publication, but I do think it would help broaden the appeal of the work.

Response: We agree with the referee that there is a wider biogeochemical story to the formation of Hells Bells, but suggest to wait for more data which are not yet available. We have taken additional samples for genome analysis of water and the surface of Hells Bells which are currently under evaluation and we are hopeful that these results will lead to a wider field of interpretation.

Referee: The second comment is a bit more substantive. I was not convinced that Hurricane activity would be likely to be responsible for sustained upward migration of the mixing zone necessary for Hells Bells to precipitate in areas above the modern redoxcline. The authors might explore droughts, which are well-documented in the Holocene (see Hodell et al., 2001, reference included in comments below), as an alternative process. Droughts would thin the freshwater lens and elevate the mixing zone for prolonged periods of time needed for the slow precipitation of calcite mentioned in the paper to form large Hells Bells features.

Response: We agree that the well documented droughts for the Yucatán Peninsula and a subsequently thinner fresh water layer brought up the halocline and certainly had its effect on Hells Bells formation. We added this point in the MS on page 24 lines 16–18.

We are currently working on the Hells Bells from different angles and produced an amount of data that cannot be shown in one manuscript only. The hypothesis of a recharge-driven upward migration of the halocline was developed integrating all information available, including some that is not shown here. Manuscripts presenting this data are currently in preparation.

The main argument why we did not consider droughts as a mechanism of halocline elevation is U/TH age-dating on Hells Bells specimens of different water depths (MS currently in preparation) show approximately identical young ages (∼150 a) at the low-ermost crystal tips (1-2 mm) of the Bells. There is even a weak trend of the youngest samples found in the lowest water depths and the oldest samples found in greater water depths. This makes droughts or prolonged periods of time with an elevated halocline as the sole mechanism for the elevation of the halocline unlikely because this should be reflected in an age-zonation of the Hells Bells.

We are aware of the fact that the described behavior of the halocline in response to recharge is counter-intuitive and that the cited reference (Escolero et al., 2007) does not explain sufficiently why the halocline should be elevated as a response to recharge events. Yet, such a behavior is well documented by their results and additionally we discovered an online available thesis investigating dynamics of the coastal Karst aquifer around Merida that reports a similar halocline elevation in response to recharge events (Heise, 2013). In general, however, there is a scarcity of studies addressing aquifer responses to recharge events in the area and there is only one other that we know of, which is Kovacs et al., (2017) and this study does not have information about the halocline elevation as it only captured data from the fresh water layer.

We try to fill this gap of data and knowledge as we are currently gathering continuous data on the halocline elevation at El Zapote cenote. Preliminary results of one year of measurement are very interesting and reveal a complex hydraulic behavior. Generally, the halocline elevation follows the elevation of the fresh water lens which seems to be controlled by the mean tidal sea level throughout the year which leads to a variation of ∼0.5 m of the halocline elevation. Furthermore, in general, if the freshwater lens gets thinner (water level of the fresh water drops), the halocline rises (determined by increasing conductivity of the data logger in the halocline) and vice versa. However, we detected several short-termed events (∼ 1 day) of an abrupt elevation of the fresh water accompanied by an elevation of the halocline as a response to precipitation/recharge

(Fig .1). Subsequently, the halocline slowly drops as expected from an increase of the fresh water lens due to precipitation. So far, there has not been a major precipitation event (hurricane) within the recorded available data (from Dec 2017- Dec 2018), but the observed effect of an initial halocline rise in response to recharge should increase in scale and time with an increase of the precipitation/recharge event.

Caption to Fig. 1: Preliminary results of data loggers from El Zapote cenote showing two examples of halocline elevation events. Water level was measured in the fresh water layer in 15 min intervals with a temperature and depth Logger (TD-Logger) in ∼6 m water depth. In the halocline conductivity, temperature and depth was measured in ∼38 m water depth also in 15 min intervals (CTD-Logger). The plots show the deviation of the respective elevation with respect to the start of measurement in December 2017.The halocline elevation is calculated from the conductivity values through a linear regression of the electrical conductivity within the interval of 37–40 m water depth. Note the periodic data noise in the halocline elevation, which always occur between 10–16h almost every day. This is most likely caused by diving and or by jumping activities into the cenote. Both cause small turbulences in the stratified water column.

Referee's minor comments:

Referee: Abstract Line 20: Given that this is a sinkhole, I'm uncomfortable with the author's description of this chamber as lightless in the absence of measurements of light intensity over annual timescales. While it may appear dark to divers, there may be some "light" that still makes it to the chamber during some portions of the year.

Response: You are right, we cannot rule out that there might be some portions of light at some time of the year. We even tried to measure the amount of light with water depth at the cenote around noon in December 2017. We found no measurable light below ∼16 m water depth (<11 Lux). However, we cannot exclude that small portions of light may penetrate deeper into the water body for some daytime and season.

To meet the referee's comment we replaced "lightless" with "dark" in Abstract Line 20.
Referee: Abstract Line 32: It would be nice if the authors could include information about why Hell's Bells are not more widely distributed in coastal aquifers in general (and the Yucatan more specifically) in the abstract. Improved information and word economy in the preceding sections should create space for this insertion.

Response: To meet the referee's suggestion we restructured the sentences and added some Information about the prerequisites for Hells Bells forrmation.

Additionally, we are addressing that issue in a MS (in prep.) in which we compare several cenotes with Hells Bells and cenote Angelita. One of the aspects of this study will be the spatial distribution of cenotes with Hells Bells

Referee: Page 2 Line 1: I've been working on karst research problems for almost 20 years and have never heard the term "pending speleothems." That doesn't mean the term does not exist, but it probably means it is not common.

Response: "Pending" was used to refer to the group of pendant speleothemes in this context, however we followed the referee's concerns and deleted "Pending" and changed the according sentence into "Speleothems, such as stalactites or dripstones, result from physicochemical. . .".

Referee: Page 3 lines 18-19: I don't understand why a recharge event would elevate the freshwater lens. It should depress it, as recharge would increase the thickness of the lens:. The manuscript that was cited wasn't much help in explaining the process either.

Response: We try to be more cautious around the halocline dynamics throughout the MS and therefore change the lines 18-19 on Page 3 to "Although Moore et al. (1992) and Stoessell et al. (1993) report that the thickness of the freshwater lens does not vary significantly between seasons or on a yearly basis, local and short-termed variations are possible and were reported by Escolero et al. (2007), who documented a significant halocline elevation of up to 17.5 m in between two measurements in the years 2000

and 2003."

Referee: Page 10, Line 3: I suggest "oxygenated" or "aerobic" as an alternative to "oxygenized"

Response: Changed to "oxygenated".

Referee: Page 11, Figure 3: The symbols change in panel C but there is no corresponding legend. I'm guessing that the circles are still winch collected and the squares are diver collected, but it would be helpful if the symbols remained constant across each panel.

Response: We agree that this the symbols are misguiding. In panel c the squares and circles stand for different parameters and it is not distinguished between samples taken by divers and those taken with the Niskin bottle. We will address this issued in the final revision once we received all referee comments.

Referee: Page 17 line 9: In this context, "though" is being used to contrast ideas and should not begin a sentence. I suggest restructuring this sentence and the previous one.

Response: Both sentences were restructured as suggested.

Referee: Page 18 line 9: Bells should be plural.

Response: Corrected as suggested.

Referee: Page 18 lines 13-15: I can agree that sulfide oxidation drives calcite precipitation but I'm still clear on why these features are only found here and not in any of the other sulfidic halocline caves in the Yucatan (or elsewhere), but hopefully this is discussed later in the text: : :..

Response: It is discussed later in the MS.

Referee: Page 19 lines 23-24: I find these sentences super confusing. Can you please

rephrase them?

Response: The passage was rephrased as recommended.

Referee: Page 19 line 28: change to "debris mound"

Response: Changed as recommended to debris mound throughout the text and also in the legend of Fig. 3.

Referee: Page 20 lines 6-8: I think the authors might want to make this section more clear by saying that methanogenesis in the presence of sulfate isn't expected, but that the methanogenesis could be occurring in the sediments where sulfate is reasonably expected to be depleted. I would then recommend they cite a manuscript or two from the marine sediment pore water research community to demonstrate this depletion is common. The authors' interpretation is also supported by the concentration profile, a point which should be emphasized in the discussion.

Response: We added "...and methanogenesis is not expected in the presence of sulfate, ..." in order to be more clear in this section. Also we followed the referee's advice and cited supporting literature of pore water profiles.

It is not clear for the authors what the referee meant with "The authors' interpretation is also supported by the concentration profile, a point which should be emphasized in the discussion". We do not think that the concentration profile in the water column reflects the concentration profile in the sediments, if that's what was meant by the referee. Instead, the concentration profile of the species that are released from organic matter decay are a consequence of the 2 sinks of the system, which are the redoxcline above and an advective depletion through a hydraulic connection somewhere below the cenote (<60 m water depth).

Referee: Page 20, line 13: It might be more clear to say that Fe is only present in low concentrations in limestone and there are limited/no sources of siliciclastic materials that could contribute iron in this part of the Yucatan. Fe-oxides are common in many

karst areas where the Fe is derived from weathering of siliciclastics.

Response: This part was changed as recommended.

Referee: Page 24, Section 4.3.1 (entire section) – I find the role of Hurricanes in elevating the halocline to be a highly implausible explanation for the vertical zonation of the Hells Bells features. Hurricanes are infrequent and the "piston induced oscillation" effect described by the authors would be short lived. I find it hard to believe that any potential upward displacement by this mechanism would be of sufficient duration to allow for the precipitation of large masses of carbonate photographed above the modern redoxcline. I suggest that a more plausible scenario is that periods of drought (which are well-documented over the Holocence) lowered the water table and increased the elevation of the redoxcline for sustained periods of time (see Hodell, D.A., Brenner, M., Curtis, J.H. and Guilderson, T., 2001. Solar forcing of drought frequency in the Maya lowlands. Science, 292(5520), pp.1367-1370.)

Response: We already referred to this comment in detail above.

Referee: Page 28: Section 4.4 prerequisites for formation of Hells Bells. Many cenotes in the Yucatan meet these criteria yet lack Hells Bells. I wonder if one of the more unique aspects of El Zapote is the relatively thick mixing zone, which the authors highlight in their discussion. Many of the haloclines are much thinner, typically due to high flux of fresh or saline water (though I was surprised by how thin the halocline at Cenote Angelita was in the supporting literature – I wouldn't have guessed there was much flow in that location). I suppose many of the haloclines with high fluxes of organic material also have much greater light penetration (such as El Pit and Cenote Angelita), so the location of the mixing zone in a dark cenote (I still hesitate to say that this is completely dark) is also likely important.

Response: Yes, we are also convinced that the thickness of the halocline is an important factor as a thick halocline results from highly stagnant conditions. These have to be met in order to create an oxygen deficient redoxcline where anaerobic protonconsuming sulfide oxidation can occur. We investigated 3 cenotes with Hells Bells and found that all show a very thick halocline, some even thicker than that at El Zapote, and found that the characteristics of the redoxcline are very different to those of cenotes with a thinner halocline, which matches most of the studied cenotes at the Yucatán Peninsula (MS in preparation).

It seems however, that light intensity is not an important factor because Hells Bells occurrence ranges from close to absolute dark cenotes (i.e. cenote Holbox, which has only one tiny opening and is occupied by bats, that probably provide the organic material) to potentially "bright" cenotes (Cenote Tortugas, which has an opening of approximately the same size as cenote Angelita).

Referee: Page 27, line 19-20: I still wonder if the general orientation of all of the speleothems towards the center of the cenote indicates some phototaxic behavior. It would certainly be the simplest explanation and I would encourage the authors to measure light intensity over an annual cycle to see if perhaps small amounts of light are penetrating (not having visited this particular site, I'm unsure how likely this is, but the Cenote profile included in Fig 3, coupled with the fact that the cenote is sufficiently large in diameter to permit cavern tours, suggests that the entrance may be sufficiently large to allow some small amounts of light during some parts of the day/year).

Response: Yes, we cannot rule out that some phototaxic behavior influences Hells Bells formation because small amounts of light may penetrate the cenote. We agree that this would be the simplest explanation. However, we tried to find an explanation for the observed orientation of the speleothems that is consistent with our results and the hypothesis on the biologically-induced mechanism for calcite precipitation. The chemolithoautotrophy within the redoxcline/turbid layer (which is another barrier for light penetration) is the driving agent for Hells Bells formation, thus we concluded that Hells Bells orientate accordingly to the availability of the "food" of those chemolithoautotrophic organisms. To measure the amount of light over time within the cenote would be a good approach though, in order to clarify this particular problem.

References: Escolero, O., Marin, L. E., Domínguez-Mariani, E. and Torres-Onofre, S.: Dynamic of the freshwater – saltwater interface in a karstic aquifer under extraordinary recharge action : the Merida Yucatan case study, Environ. Geol., 51, 719–723, doi:10.1007/s00254-006-0383-1, 2007. Heise, L.: Dynamics of the coastal Karst aquifer in northern Yucatán Peninsula. [online] Available from: http://ninive.uaslp.mx/jspui/handle/i/3692, 2013. Kovacs, S. E., Reinhardt, E. G., Stastna, M., Coutino, A., Werner, C., Collins, S. V., Devos, F. and Le Maillot, C.: Hurricane Ingrid and Tropical Storm Hanna's effects on the salinity of the coastal aquifer, Quintana Roo, Mexico, J. Hydrol., 551, 703–714, doi:10.1016/j.jhydrol.2017.02.024, 2017. Moore, Y. H., Stoessell, R. K. and Easley, D. H.: Fresh-Water/Sea-Water relationship within a Ground-Water flow system northeastern coast of the yucatan, Ground Water, 30(3), 343–350, 1992. Stoessell, R. K., Moore, Y. H. and Coke, J. G.: The Occurrence and Effect of Sulfate Reduction and Sulfide Oxidation on Coastal Limestone Dissolution in Yucatan Cenotes, Ground Water, 31(4), 566–575, 1993.

[Figure]

**Fig. 1.** Preliminary results of data loggers from El Zapote cenote showing two examples of halocline elevation events. (Full caption is given on page C4)

---

## Referee Comment (RC2) · Anonymous Referee #2 · 26 Feb 2019

General comment:

This paper presents detailed hydrogeochemical and geochemical analyses of the water column and the so-called Hells Bells formed in a cenote on the Yucatan Peninsula, Mexico, in order to determine the processes leading to the development of the Hells Bells. This is an interesting topic because these submerged speleothems are a unique feature suggesting that their growth is only possible in case of very specific conditions.

The paper is, in general, well written, the results are clearly presented and the developed hypotheses are sound and justified by the data. The text is here and there a bit lengthy (in particular, the conditions leading to the formation of the Hells Bells are repeated several times), but this is not a major issue.

I have two (moderate to major) general comments that should be addressed by the authors prior to publication. The first – and more serious - comment is related to the changes in the depth of the halocline, which are considered as the reason resulting in growth of Hells Bells at different water depths or of large bells over a longer time. The authors develop a hypothesis that the depth of the halocline is related to fresh water recharge at the surface and even suggest a potential relationship with the occurrence of hurricanes. However, the time scale of the growth of the Hells Bells is not discussed with sufficient detail in the paper. Considering the enormous size of at least some of the bells, it is hard to believe that these should have developed due to seasonal or episodic changes in the depth of the halocline. I would rather believe that this requires a long-term shift in the depth of the halocline, for instance over several thousand years during the Holocene. I had a quick look at the previous paper of the same group (Stinnesbeck et al., 2017b), which presented a few U-series data and reported growth rates of ca. 10-100 $\mu$m/a. In case of such slow growth rates, it is hard to believe that a short-term decrease in the depth of the halocline due to a recharge event would have a visible effect. In contrast, growth of a really large bell, requires slow and progressive changes in the (mean) depth of the halocline. The U-series ages ranging from a few hundred to a few thousand years reported by the previous paper, actually seem to confirm this. Thus, the authors should ideally present many more U-series data trying to resolve the timing and duration of the growth of the Hells Bells. If this is technically impossible or beyond the scope of the paper, they could also present data from a second campaign, probably shortly after a major recharge event. As far as I understand, these measurements are currently conducted (p. 26, line 30ff.) and could easily be included in a revised paper. If the authors do not want to include additional data (neither U-series, nor elevation data of the halocline), they must clearly address this issue in the revised paper and

critically discuss the time-scales of the dynamics of the halocline and the growth rate of the Hells Bells.

My second point concerns the typical bell-shape of the Hells Bells. In section 4.3.2, the authors discuss several processes, but none of the them – as far as I understood - explains the "conically divergent" (p. 27, line 14) shape of the bells. This issue should either be addressed more clearly, or it should be stated in the MS that the processes discussed in the text cannot completely explain the typical shape of the bells.

In summary, although the paper is generally very interesting and well written, and the data clearly deserve publication, I can only recommend publication in Biogeosciences after revision. Below, I list a few additional, more detailed comments.

Detailed comments:

Page 6, line 9: Why are only data from a single campaign reported? In particular considering the important aspect of the dynamics of the halocline (see above), it would be much better to provide at least a few data from an additional campaign conducted shortly after a major recharge event (hurricane).

P. 6, line 20ff: "Due to increasing sulfide concentrations in water depths below the turbid layer and interaction of sulfide with the Ag/Cl pH electrode, a shift of pH of up to 0.2 pH units towards higher values was observed when comparing the pH logs of the way down with the pH logs of the way up (Fig. S1). This shift is dependent on the exposure time of the electrode and the respective sulfide concentrations and could not be quantified nor corrected for." May it be possible to quantify the effect in the laboratory by increasing the sulfide content of a test-solution?

P. 12, line 19ff.: "However, SI values calculated for the halocline suffer from the overestimated pH readings in the extremely sulfidic water of the halocline and are therefore not considered." This is a pity because supersaturation with respect to calcite within the halocline is the hypothesis presented to explain the growth of the Hells Bells. Thus,

it would be really good to estimate the effect on the pH values (see above) and the resulting effect on SI. This information is essential for the validity of the presented hypothesis.

P. 13, line 5: "... from X to Y ..." Is this an artefact from a previous version of the paper?

P. 13, line 5: Fig. 5 should be Fig. 4.

P. 14, line 6: Figs 6 should be Figs. 5.

P. 16, line 6ff.: Please state here that the samples were collected from "several" specimens. This information is important. In addition, it is (again) problematic that no dating is provided. Then, the data would not be related to the "presumably youngest part" of the bells, but the age of the samples could be precisely determined.

P. 16, line 11: "soluble" should be "insoluble"?

P. 16, line 21: d13Ccalcite values show a strong negative (not a positive) correlation with Sr/Ca and Ba/Ca if I correctly read Fig. 6.

P. 16, line 25: In my opinion, the offset between the calculated and the measured d13C value of the HCO3 (2-5 permille) is substantial. Thus, speaking of "slightly lower" values is not correct.

Section 4: The introductory section could be deleted to make the paper more concise.

Section 4.2: See above. The introductory section could be deleted to make the paper more concise.

P. 22, line 7: Please provide a reference for reaction (R1).

P. 22, line 21: Please define "SD-OM".

P. 22, line 27: "As organisms usually prefer to metabolize 12C (it takes less energy to break the 12C bond instead of 13C) they effectively consume HCO3- with lower d13C

values, which subsequently results in higher d13CHCO3- values in the remaining dissolved inorganic carbon" It is true that the organisms preferentially metabolise 12CO2, but they do not "effectively consume HCO3- with lower d13C values". The preferential consume of 12CO2 (and the related increase in the d13C value of the CO2) leads to chemical and isotopic reactions resulting in conversion of HCO3 into CO2 and an increase in the d13C value of the HCO3 reservoir.

P. 23, line 2: "It was shown before that Hells Bells form within the freshwater indicated by d234Uinitial values of 16–25 ‰ of the Hells Bells calcite (Stinnesbeck et al., 2017b)." Please explain this statement. Why do these values suggest precipitation within freshwater? Due to the non-marine d234U value (lower than 150 permille)? Or has the d234U value of the water in the cenote been determined (at different depths)? Actually, freshwater often has higher d234U values than seawater . . .

P. 24, line 8: "Hells Bells formed in modern to historic times . . ." How do you know that? Is this statement based on the few U-series ages reported by Stinnesbeck et al. (2017b)? It may very well be possible that there are much older specimens in the same cenote.

P. 24, line 13ff.: "Therefore, we propose that growth of Hells Bells is a non-permanent episodic process which majorly depends on a highly variable halocline elevation in the cenote (Fig 10)." See my major comment above. The probably very different growth rate of the bells and the seasonal to episodic dynamics of the depth of the halocline need to be discussed in detail.

P. 24, line 16ff.: "Extraordinary recharge events (e.g. hurricanes) must have a significant effect on the depth position of this layer . . ." See above. Even if this is the case, it is not clear whether these episodic changes would be recorded by the slowly growing Hells Bells.

---

## Author Comment (AC2) · 8 Mar 2019

Author's Response to the Interactive comment on "Subaqueous speleothems (Hells Bells) formed by the interplay of pelagic redoxcline biogeochemistry and specific hydraulic conditions in the El Zapote sinkhole, Yucatán Peninsula, Mexico" by Simon Michael Ritter et al.

The authors are very grateful for the time and effort taken for reviewing the manuscript

in detail and appreciate the constructive comments and suggestions by anonymous referee #2. We addressed all issues raised by the referee in a point by point response below.

Referee's general comment: This paper presents detailed hydrogeochemical and geochemical analyses of the water column and the so-called Hells Bells formed in a cenote on the Yucatan Peninsula, Mexico, in order to determine the processes leading to the development of the Hells Bells. This is an interesting topic because these submerged speleothems are a unique feature suggesting that their growth is only possible in case of very specific conditions.

The paper is, in general, well written, the results are clearly presented and the developed hypotheses are sound and justified by the data. The text is here and there a bit lengthy (in particular, the conditions leading to the formation of the Hells Bells are repeated several times), but this is not a major issue.

I have two (moderate to major) general comments that should be addressed by the authors prior to publication. The first – and more serious - comment is related to the changes in the depth of the halocline, which are considered as the reason resulting in growth of Hells Bells at different water depths or of large bells over a longer time. The authors develop a hypothesis that the depth of the halocline is related to fresh water recharge at the surface and even suggest a potential relationship with the occurrence of hurricanes. However, the time scale of the growth of the Hells Bells is not discussed with sufficient detail in the paper. Considering the enormous size of at least some of the bells, it is hard to believe that these should have developed due to seasonal or episodic changes in the depth of the halocline. I would rather believe that this requires a longterm shift in the depth of the halocline, for instance over several thousand years during the Holocene. I had a quick look at the previous paper of the same group (Stinnesbeck et al., 2017b), which presented a few U-series data and reported growth rates of ca. 10-100 $\mu$m/a. In case of such slow growth rates, it is hard to believe that a short-term decrease in the depth of the halocline due to a recharge event

would have a visible effect. In contrast, growth of a really large bell, requires slow and progressive changes in the (mean) depth of the halocline. The U-series ages ranging from a few hundred to a few thousand years reported by the previous paper, actually seem to confirm this. Thus, the authors should ideally present many more U-series data trying to resolve the timing and duration of the growth of the Hells Bells. If this is technically impossible or beyond the scope of the paper, they could also present data from a second campaign, probably shortly after a major recharge event. As far as I understand, these measurements are currently conducted (p. 26, line 30ff.) and could easily be included in a revised paper. If the authors do not want to include additional data (neither U-series, nor elevation data of the halocline), they must clearly address this issue in the revised paper and critically discuss the time-scales of the dynamics of the halocline and the growth rates of the Hells Bells.

Response: We fully agree that growth rates should be discussed in the MS and followed the referee's suggestion by critically discussing and estimating Hells Bells growth rates. We address this issue in a new chapter in the revised MS (Section: "4.3.1 Calcite precipitation rates"). Concerning the presentation of additional data, we are currently working on the Hells Bells from different angles and produced an amount of data that cannot be shown in one manuscript only. Manuscripts, also addressing age-dating of Hells Bells, are currently in preparation. We conducted a second campaign which confirmed the data presented in the MS, but the major goal of second campaign was to study other cenotes showing Hells Bells and compare them to a cenote without Hells Bells. We are currently preparing a manuscript presenting these data. Additionally, as the dynamic halocline elevation was also of major concern for Anonymous referee #1 and we already addressed a lot of the raised issues and we would therefore also like to refer to the Author's response AC1 (https://www.biogeosciences-discuss.net/bg-2018-520/#discussion).

Referee: My second point concerns the typical bell-shape of the Hells Bells. In section 4.3.2, the authors discuss several processes, but none of the them – as far as I

understood – explains the "conically divergent" (p. 27, line 14) shape of the bells. This issue should either be addressed more clearly, or it should be stated in the MS that the processes discussed in the text cannot completely explain the typical shape of the bells.

Response: Absolutely correct, we did not comment on the conical divergent shape of Hells Bells but of course we thought about that, too. We think that the conical shape of Hells Bells and their downward divergence could be the macroscopic expression of the calcite crystal's microscopic features. We observed two major phases of calcite crystals, blocky or mosaic and elongated doog-dooth calcites. The latter are arranged in a botryoidal structure which strongly resembles the shape of the larger Hells Bells structures. We added this observation into the discussion section 4.3.2 and added a thin section photograph showing such features to the supplement (Fig. S4 in the revised Supplement). Preferential or faster growth on the outer edges of Bells, especially the edges pointing towards the center of the cenotes (this is described in detail in section 4.3.2) may lead to the hollowness because once these outer edges of a specimen have grown slightly deeper that the rest of the specimen, then the net growth of these parts will be higher because it is more frequently reached by the zone of calcite precipitation. This would ultimately lead into the observed hollowness of the larger specimen of Hells Bells. We also integrated these thoughts on the Hells Bells features into the manuscript in section 4.3.2

Referee: In summary, although the paper is generally very interesting and well written, and the data clearly deserve publication, I can only recommend publication in Biogeosciences after revision. Below, I list a few additional, more detailed comments.

Referee's detailed comments:

Referee: Page 6, line 9: Why are only data from a single campaign reported? In particular considering the important aspect of the dynamics of the halocline (see above), it would be much better to provide at least a few data from an additional campaign

conducted shortly after a major recharge event (hurricane).

Response: We installed data loggers in the cenote for permanent observation of the halocline elevation and water level. This data is so far (~1 year) very promising, however there was no major precipitation event like a since the start of the measurement in December 2017. We elaborated this issue in our response to Referee comment #1 (AC1 on pages C2–C3 https://www.biogeosciences-discuss.net/bg-2018-520/#discussion). Furthermore, the main goal of this study is to present detailed hydrogeochemical results in order to develop a hypothesis on the mechanism for subaqueous calcite precipitation. The aspect a halocline elevation is a direct consequence of the conclusion that calcite most likely precipitates within the narrow redoxcline. In the revised MS we are more careful in discussing the halocline dynamics and clearly state that this is not supported by data, yet.

Referee: P. 6, line 20ff: "Due to increasing sulfide concentrations in water depths below the turbid layer and interaction of sulfide with the Ag/Cl pH electrode, a shift of pH of up to 0.2 pH units towards higher values was observed when comparing the pH logs of the way down with the pH logs of the way up (Fig. S1). This shift is dependent on the exposure time of the electrode and the respective sulfide concentrations and could not be quantified nor corrected for." May it be possible to quantify the effect in the laboratory by increasing the sulfide content of a test-solution?

Response: Please see the following response below.

Referee: P. 12, line 19ff.: "However, SI values calculated for the halocline suffer from the overestimated pH readings in the extremely sulfidic water of the halocline and are therefore not considered." This is a pity because supersaturation with respect to calcite within the halocline is the hypothesis presented to explain the growth of the Hells Bells. Thus, it would be really good to estimate the effect on the pH values (see above) and the resulting effect on SI. This information is essential for the validity of the presented hypothesis.

Response: Unfortunately, quantifying the pH shift is not possible because the sonde that we used was a rental sonde. However, we did this campaign again at El Zapote cenote in 2018 with the exact same type of sonde but a new one. The pH values were identical to the ones presented in the MS except for the values below the redoxcline which were significantly lower at its minimum values in around 42 m water depth (up to ∼0.25 pH units). We fully agree that this is essential for the validity of our hypothesis and therefore addressed this issue by including pH and SI calcite data from the sampling campaign in June 2018 into Supplement Fig. S2. We refer to this information in the method section and section "3.1.1 Calcite Saturation". This allows now to clearly show that calcite is oversaturated in the redoxcline and undersaturated in the halocline indicating calcite dissolution.

Referee: P. 13, line 5: ": : : from X to Y : : :" Is this an artefact from a previous version of the paper?

Response: Yes, thank you. It was corrected in the MS with the according values.

Referee: P. 13, line 5: Fig. 5 should be Fig. 4.

Response: Corrected as suggested.

Referee: P. 14, line 6: Figs 6 should be Figs. 5.

Response: Corrected as suggested.

Referee: P. 16, line 6ff.: Please state here that the samples were collected from "several" specimens. This information is important. In addition, it is (again) problematic that no dating is provided. Then, the data would not be related to the "presumably youngest part" of the bells, but the age of the samples could be precisely determined.

Response: Corrected as suggested. We are currently working on a Manuscript with the focus on age-dating of the Hells Bells.

Referee: P. 16, line 11: "soluble" should be "insoluble"?

Response: Yes, of course. Thank you. Corrected as suggested.

Referee: P. 16, line 21: d13Ccalcite values show a strong negative (not a positive) correlation with Sr/Ca and Ba/Ca if I correctly read Fig. 6.

Response: You are right. Corrected as suggested.

Referee: P. 16, line 25: In my opinion, the offset between the calculated and the measured d13C value of the HCO3 (2-5 permille) is substantial. Thus, speaking of "slightly lower" values is not correct.

Response: We agree and deleted "slightly", now saying that the s13C value of the measured HCO3 is lower than the calculated d13Ceq HCO3 vlaue.

Referee: Section 4: The introductory section could be deleted to make the paper more concise.

Response: We followed the referee's suggestions and deleted this section.

Referee: Section 4.2: See above. The introductory section could be deleted to make the paper more concise.

Response: We followed the referee's suggestions and deleted this section.

Referee: P. 22, line 7: Please provide a reference for reaction (R1).

Response: Reaction R1 was developed by the authors. However, in order to meet the referee's comment we referenced to supporting literature that shows the process of anaerobic sulfide oxidation via nitrate ( "...could be obtained with nitrate as terminal electron acceptor producing elemental sulfur and nitrogen under the consumption of protons (e.g. Bailey et al., 2009)."

Referee: P. 22, line 21: Please define "SD-OM".

Response: SD-OM is already defined in the MS above ("...could be obtained with nitrate as terminal electron acceptor producing elemental sulfur and nitrogen under the

consumption of protons (e.g. Bailey et al., 2009)."

Referee: P. 22, line 27: "As organisms usually prefer to metabolize 12C (it takes less energy to break the 12C bond instead of 13C) they effectively consume HCO3- with lower d13C values, which subsequently results in higher d13CHCO3- values in the remaining dissolved inorganic carbon" It is true that the organisms preferentially metabolise 12CO2, but they do not "effectively consume HCO3- with lower d13C values". The preferential consume of 12CO2 (and the related increase in the d13C value of the CO2) leads to chemical and isotopic reactions resulting in conversion of HCO3 into CO2 and an increase in the d13C value of the HCO3 reservoir.

Response: We agree and deleted "effectively consume HCO3- with lower d13C values" . We changed the according sentence to "Organisms usually prefer to metabolize 12C (it takes less energy to break the 12C bond instead of 13C), which results in higher ïĄď13C-HCO3- values in the remaining dissolved inorganic carbon."

Referee: P. 23, line 2: "It was shown before that Hells Bells form within the freshwater indicated by d234Uinitial values of 16–25 ‰ of the Hells Bells calcite (Stinnesbeck et al., 2017b)." Please explain this statement. Why do these values suggest precipitation within freshwater? Due to the non-marine d234U value (lower than 150 permille)? Or has the d234U value of the water in the cenote been determined (at different depths)? Actually, freshwater often has higher d234U values than seawater

Response: We agree that we cannot state this without showing the d234U values over the depth range of the water body. We removed the argument of d234U and changed the sentence to "It was suspected before that Hells Bells form within the freshwater body of El Zapote cenote (Stinnesbeck et al., 2017b)."

Referee: P. 24, line 8: "Hells Bells formed in modern to historic times : : :" How do you know that? Is this statement based on the few U-series ages reported by Stinnesbeck et al. (2017b)? It may very well be possible that there are much older specimens in the same cenote.

Response: This statement is based on the U-series measurements and the 14C age of the tree stem which is covered by small specimen of Hells Bells. We agree that there should be much older specimen considering their enormous size. We changed the sentence to "Hells Bells formed in modern to at least historic times and occur…" by inserting "to at least historic times" in order to meet the referee's critique.

Referee: P. 24, line 13ff.: "Therefore, we propose that growth of Hells Bells is a non-permanent episodic process which majorly depends on a highly variable halocline elevation in the cenote (Fig 10)." See my major comment above. The probably very different growth rate of the bells and the seasonal to episodic dynamics of the depth of the halocline need to be discussed in detail.

Response: We fully agree and added a detailed discussion on growth rates to the revised MS. However, although we are convinced that the halocline dynamic is an interesting and upcoming research topic we believe that a detailed discussion on the dynamic elevation of the depth of the halocline is beyond the scope of the MS.

Referee: P. 24, line 16ff.: "Extraordinary recharge events (e.g. hurricanes) must have a significant effect on the depth position of this layer : : :" See above. Even if this is the case, it is not clear whether these episodic changes would be recorded by the slowly growing Hells Bells.

Response: The detailed discussion on growth rates in the revised MS demonstrates that short-term episodic events could be recorded in Hells Bells growth.

―――――――――――――――――――

---

## Referee Report (RR1)

Review of the revised manuscript "Subaqueous speleothems (Hells Bells) formed by the interplay of pelagic redoxcline biogeochemistry and specific hydraulic conditions in the El Zapote sinkhole, Yucatán Peninsula, Mexico" by Ritter et al., submitted to Biogeosciences

General comment:

I thank the authors for their detailed response to my comments.

My major concern (and obviously of the other reviewer as well) was the hypothesis of the authors that the growth of the Hells Bells is controlled by the depth of the halocline and the redoxcline, which, in turn, are related to recharge. Based on this hypothesis, the authors suggested episodic rather than continuous growth of the Hells Bells (at different depth) and even proposed a relationship with extreme events, such as hurricanes.

In my comment, I stated: "Considering the enormous size of at least some of the bells, it is hard to believe that these should have developed due to seasonal or episodic changes in the depth of the halocline. I would rather believe that this requires a long-term shift in the depth of the halocline, for instance over several thousand years during the Holocene. I had a quick look at the previous paper of the same group (Stinnesbeck et al., 2017b), which presented a few U-series data and reported growth rates of ca. 10-100 µm/a. In case of such slow growth rates, it is hard to believe that a short-term decrease in the depth of the halocline due to a recharge event would have a visible effect. In contrast, growth of a really large bell, requires slow and progressive changes in the (mean) depth of the halocline. The U-series ages ranging from a few hundred to a few thousand years reported by the previous paper, actually seem to confirm this."

I still think that the only way to test this hypothesis would be systematic U-series dating of several bells from various depths, which requires an enormous number of U-series ages. As far as I understand, this work is currently in progress, and the authors do not want to include these data in the current MS. This is OK, in particular in times where each PhD thesis must consist of several papers. It is still a bit unfortunate, however, for the reader of the current MS because the authors' response to the reviews makes clear that their interpretations are – at least partly – based on these additional data.

In their revised MS, the authors estimate the potential growth rates of the Hells Bells (I very much appreciate that) and obtain results in agreement with their preliminary U-series data. This shows that the potential growth rates of the bells are in the range of a few hundred µm/a. In addition, they show data from their loggers suggesting changes in the range of 10-20 cm to recharge events. Finally, they state in their reply that " … The main argument why we did not consider droughts as a mechanism of halocline elevation is U/TH age-dating on Hells Bells specimens of different water depths (MS currently in preparation) show approximately identical young ages (~150 a) at the lowermost crystal tips (1-2 mm) of the Bells. There is even a weak trend of the youngest samples found in the lowest water depths and the oldest samples found in greater water depths. This makes droughts or prolonged periods of time with an elevated halocline as the sole mechanism for the elevation of the halocline unlikely because this should be reflected in an age-zonation of the Hells Bells."

Considering all these points, I tend to agree that short-term recharge events may have the potential to change the depth of the halocline (even if not in the range of several meters) and result in episodic growth of Hells Bells at different depths. Even if the growth rates were

much lower, this could still eventually result in large bells because you have a lot of time (thousands of years). This is comparable to a speleothem that is fed by a very slow and maybe episodic drip rate or only during a specific season of the year (e.g., winter). Growth is episodic, but you may still receive a large speleothem after sufficient time, which appears to have been grown continuously.

In summary, I tend to generally agree with the authors' hypothesis, now that I have seen the additional data. However, there are two important points, which should be added to the MS prior to publication to make the suggested process clear to reader:

1. Due to the slow growth rates of µm/a, it will not be possible to *reconstruct* the detailed episodic growth history of a single bell on the seasonal or even annual time scale by U-series dating. If the water level and the depth of the halocline fluctuates on a daily, weekly, seasonal or even annual time-scale, you will always have growth at different depths in the cenote throughout the year. Thus, a *reconstruction* of individual extreme events (e.g., hurricanes) by precise U-series dating will not be possible using the Hells Bells. It is, thus, misleading to state that extreme events, such as hurricanes, can be *recorded* by the bells. Therefore, I suggest to delete the reference to hurricanes.
2. As far as I understand, growth of the bells in the cenote is restricted to a relatively large range in depth of ca. 10 m. This seems to be a lot compared to the fluctuations of 10-20 cm observed in the logger data (Fig. 1 in the reply of the authors). Thus, larger changes in depth may be related to long-term processes (centennial to millennial scale) and minor changes to short-term events. This should be made clear in the text even if the logger data are not shown.

In summary, I recommend to accept the MS after the suggested changes have been made.

---

## Author Response (AR2)

**Author´s response to the reviews of the revised manuscript "Subaqueous speleothems (Hells Bells) formed by the interplay of pelagic redoxcline biogeochemistry and specific hydraulic conditions in the El Zapote sinkhole, Yucatán Peninsula, Mexico" by Ritter et al., submitted to Biogeosciences**

5   Dear Dr. Wajih Naqvi,

On behalf of all Co-authors, I would like to thank you and the anonymous referees for the excellent editorial handling and reviewing of our manuscript. Please find attached our point-by-point response to the comments of anonymous referees #1 and #2 to our revised manuscript and a list summarizing the relevant changes made to the revised manuscript.

Sincerely,

Simon Ritter

15   **Author´s response to the comments of anonymous referee #1:**

*Referee:*

*I'd like to reiterate from my previous review that I think the authors have done an excellent job examining the processes controlling Hell's Bells formation over a series of manuscripts. I'm looking forward to reading the additional manuscripts they mention are in preparation.*

*I have only a few general criticisms and hope to see the paper published in due course. First, the paper is now entirely too long. Second, and related to the first, at least some of the length of the manuscript can be reduced by removing the detailed descriptions of all of the different ways that the halocline can oscillate (see comments related to Page 26). I briefly expand on these points in my line-by-line comments, below.*

25   *Page 1, Line 30 – I recommend adding annual tidal variability to the list of possible causes for halocline oscillation. The authors mention this in their rebuttal, but do not include it in the manuscript.*

    Response: We added "annual tidal variability" as suggested by the referee.

*Page 2, Line 5 – Stalactites, by definition, only form by dripping water. They cannot form underwater. Please change to "speleothems" or "secondary precipitates."*

30       Response: Corrected as suggested. We also deleted the previous sentence as the information of how stalactites form is no longer needed.

*Page 2, line 12 – meter should be plural.*

Response: Corrected as suggested.

*Page 2, line 15 – if they are above the anoxic/sulfidic zone, how is the environment "toxic"?*

Response: That´s correct, "toxic" is now deleted in the manuscript.

*Page 3, line 4 – there is no plausible method of estimating unmapped cave lengths. I recommend removing the suggestion that there is more than 7,000 km of possible cave.*

Response: Corrected as suggested.

*Page 3, line 6 – Karst cave is probably redundant in this context.*

Response: Corrected as suggested. We deleted "karst".

*Page 3, line 10 – recommend changing "seawater" to "saline water" as the geochemical composition of some of this water is distinct from seawater.*

Response: Corrected as suggested.

*Page 3, line 11-14 – the halocline has been proposed to be an area of carbonate mineral undersaturation on the basis of numerical models and geomorphology, however, geochemical studies rarely find undersaturation due to mixing (which is the point of the Gulley et al. 2016 paper that is cited here as evidence of undersaturation in haloclines).*

Response: We are aware that the reason of the undersaturation with respect to calcite in the halocline is discussed controversial in the literature. Therefore, we avoided going into detail on the origin of the undersaturation (e.g. mixing dissolution or dissolution due to elevated $PCO_2$ derived from microbial decomposition of Corg) and only state that the halocline "…is usually characterized by undersaturation with respect to $CaCO_3$, leading to cave formation and conduit enlargement in the coastal carbonate aquifer (Back et al., 1986; Gulley et al., 2016; Mylroie and Carew, 1990; Smart et al., 2006)" in lines 12–14 on page 3 in the revised manuscript.

Thus, in our opinion, there is no need of a change to the manuscript due to the issue raised by the referee.

*Page 6, line 10 – change to tourist.*

Response: Corrected as suggested.

*Page 6 – line 23. The hysteresis observed in pH is clearly shown in the graphs, but I hesitate to conclude that the authors can interpret those results to indicate pH values should be lower in the saline water. Their pH profiles show pH values of ~6.8. Lower pH values of 6.6 seem unlikely considering the buffering capacity of seawater (Ben-Yaakov, 1973)*

Response: We measured lower pH Values of ~6.6 at El Zapote in June 2018 with a new sensor which supports the interpretation that the pH values in the halocline are likely to be lower than shown in the manuscript (shown in Fig. S2). Such low pH values (down to 6.0) were also reported for other deep stratified cenotes of the YP with dominant sulfate reduction like cenote Xcolac (Socki et. al, 2002). Additionally, the geochemical conditions in the water body of the cenote, especially in the halocline are different to that of porewater in sea sediments.

Therefore, we are convinced that the presented pH values in the revised manuscript are reliable.

*Page 6 – line 33-34 – how were the glass vials used to sample water underwater? Were the vials empty and then opened (pressure makes this unlikely). Were the vials pre-filled with water and then purged with breathing gas? If so, how many purges were used?*

Response: The vials were transported open and filled with water in bags by the divers. At the desired sampling depth, the divers replaced the water inside the vials with the surrounding water at each water depth by shaky motions and sealed the vials. We addressed this issue by adding this information to the method section in lines 32–34 on page 6 as follows: "The containers were carried open and water-filled by the divers. At the desired sample depth, the water in the containers was exchanged by surrounding water via shaky motions, sealed underwater and the water depth was noted for each sample."

*Page 23 – Fig 9 - DOC appears to spike in the turbid layer and no3 concentrations are high in freshwater above the halocline. Oxidation of organic material in sediments is important, but the data in figure 3 suggest that organic matter accumulation at the density interface in the turbid zone is also important. Is there any way to include these processes in Fig 9 and in the text referencing Fig 9?*

Response: We agree that the organic matter accumulation at the density boundary could play a role in the biogeochemical cycle. The DOC spike in the turbid layer, however, is not indicative for accumulating organic matter as DOC could also reflect metabolites in the complex chemolithotrophic biogeochemical cycle within the redoxcline or turbid layer. Thus, we are not able to determine the origin of DOC and tie it to a specific process in the turbid layer, e.g. accumulation of fine-grained organic matter at the density boundary that could be metabolized by heterotrophic bacteria. Nevertheless, we tried to represent this in Fig. 9 by the top box "minor aerobic and anaerobic heterotrophy" to suggest the possibility of these processes. However, we did not include these processes in the text referencing Fig. 9 as we intend to focus the discussion around Fig. 9 only on the processes involved in Hells Bells formation, hence the processes that lead to the observed alkaline pH-shift.

*Page 25, line 21 – replace "wide vertical zone" with "thick vertical zone" to make it more clear that the reference is to the vertical dimension.*

Response: Corrected as suggested.

*Page 26, entire section on oscillation of the halocline - I think the authors can reduce the length of this entire section by simply stating that the elevation of the halocline oscillates over multiple timescales in response to droughts, hurricanes, and annual tidal fluctuations that are superimposed upon on a longer term increase in sea level. This would allow them to dodge the complicated (and not particularly well-explained) hydraulics associated with recharge elevation of the halocline (which really doesn't add much besides length to this paper). If the authors feel this is super important, they should follow it up with a second paper with the data they mention having collected.*

Response: We agree that some parts of this section were not well explained and deleted these parts of the section as suggested by the referee. We also followed the referee´s suggestion by adding the following sentence in lines 30–31 on page 25 in the revised manuscript "Therefore, the halocline elevation can vary on multiple timescales in response

to droughts, recharge events and annual tidal fluctuations that are superimposed upon on a longer term sea level change." We then go on and briefly explain the effects of droughts and recharge events on the halocline elevation. After our opinion, the discussion on the variable halocline elevation is essential for understanding the mechanism of Hells Bells formation. Thus, we intended to make this section shorter and more focused to account for the referee´s concerns about the manuscript length.

*Page 30, Section 4.4 – In the rebuttal, the authors indicate that Hell's Bells are also found in caves with light, so the requirement for a lightless environment here is confusing. Further, I think the authors should specifically mention the requirement for a thick mixing zone. I would then go on to explain how other mixing zones that have been studied in the Yucatan differ from these requirements.*

Response: We followed the referee´s suggestion by specifically mentioning a thick halocline as a prerequisite for Hells Bells formation and deleting "lightless environment". We changed the according sentences in lines 10–11 on page 30 in the revised manuscript to "A meromictic stagnant water body indicated by a thick halocline is needed that allows for the formation of a redoxcline in which anaerobic chemolithoautotrophy prevails." and in the abstract in line 32 on page 1 to "Finally, we infer that highly stagnant conditions, i.e. a thick halocline, a low-light environment and sufficient input of organic material into a deep meromictic cenote are apparent prerequisites for Hells Bells formation."

In order to keep the manuscript focused we do not wish to include further comparisons concerning halocline thicknesses of other cenotes of the YP. We will address this particular topic in our next manuscript where we will compare several cenotes with Hells Bells to cenote Angelita, which is devoid of Hells Bells.

**Author´s response to the comments of anonymous referee #2:**

*Referee:*

*General comment:*

*I thank the authors for their detailed response to my comments.*

*My major concern (and obviously of the other reviewer as well) was the hypothesis of the authors that the growth of the Hells Bells is controlled by the depth of the halocline and the redoxcline, which, in turn, are related to recharge. Based on this hypothesis, the authors suggested episodic rather than continuous growth of the Hells Bells (at different depth) and even proposed a relationship with extreme events, such as hurricanes.*

*In my comment, I stated: "Considering the enormous size of at least some of the bells, it is hard to believe that these should have developed due to seasonal or episodic changes in the depth of the halocline. I would rather believe that this requires a long-term shift in the depth of the halocline, for instance over several thousand years during the Holocene. I had a quick look at the previous paper of the same group (Stinnesbeck et al., 2017b), which presented a few U-series data and reported growth rates of ca. 10-100 µm/a. In case of such slow growth rates, it is hard to believe that a short-term decrease in the depth of the halocline due to a recharge event would have a visible effect. In contrast, growth of a really large bell, requires slow and*

*progressive changes in the (mean) depth of the halocline. The U-series ages ranging from a few hundred to a few thousand years reported by the previous paper, actually seem to confirm this."*

*I still think that the only way to test this hypothesis would be systematic U-series dating of several bells from various depths, which requires an enormous number of U-series ages. As far as I understand, this work is currently in progress, and the*

5  *authors do not want to include these data in the current MS. This is OK, in particular in times where each PhD thesis must consist of several papers. It is still a bit unfortunate, however, for the reader of the current MS because the authors' response to the reviews makes clear that their interpretations are – at least partly – based on these additional data.*

*In their revised MS, the authors estimate the potential growth rates of the Hells Bells (I very much appreciate that) and obtain results in agreement with their preliminary U-series data. This shows that the potential growth rates of the bells are in the*

10  *range of a few hundred μm/a. In addition, they show data from their loggers suggesting changes in the range of 10-20 cm to recharge events. Finally, they state in their reply that "... The main argument why we did not consider droughts as a mechanism of halocline elevation is U/TH age-dating on Hells Bells specimens of different water depths (MS currently in preparation) show approximately identical young ages (~150 a) at the lowermost crystal tips (1-2 mm) of the Bells. There is even a weak trend of the youngest samples found in the lowest water depths and the oldest samples found in greater water*

15  *depths. This makes droughts or prolonged periods of time with an elevated halocline as the sole mechanism for the elevation of the halocline unlikely because this should be reflected in an age-zonation of the Hells Bells."*

*Considering all these points, I tend to agree that short-term recharge events may have the potential to change the depth of the halocline (even if not in the range of several meters) and result in episodic growth of Hells Bells at different depths. Even if the growth rates were much lower, this could still eventually result in large bells because you have a lot of time (thousands of*

20  *years). This is comparable to a speleothem that is fed by a very slow and maybe episodic drip rate or only during a specific season of the year (e.g., winter). Growth is episodic, but you may still receive a large speleothem after sufficient time, which appears to have been grown continuously.*

*In summary, I tend to generally agree with the authors' hypothesis, now that I have seen the additional data. However, there are two important points, which should be added to the MS prior to publication to make the suggested process clear to*

25  *reader:*

*1. Due to the slow growth rates of μm/a, it will not be possible to reconstruct the detailed episodic growth history of a single bell on the seasonal or even annual time scale by U-series dating. If the water level and the depth of the halocline fluctuates on a daily, weekly, seasonal or even annual time-scale, you will always have growth at different depths in the cenote throughout the year. Thus, a reconstruction of individual extreme events (e.g., hurricanes) by precise U-series dating will*

30  *not be possible using the Hells Bells. It is, thus, misleading to state that extreme events, such as hurricanes, can be recorded by the bells. Therefore, I suggest to delete the reference to hurricanes.*

Response: We did not intend to suggest that individual hurricanes could be identified in the Hells Bells speleothem records and apologize if it seemed so in the manuscript. We therefore followed the referee´s suggestion and deleted

the references to hurricanes from the abstract, conclusions and most parts of the manuscript and minimized referencing hurricanes to a minimum of two text passages in the revised manuscript.

*2. As far as I understand, growth of the bells in the cenote is restricted to a relatively large range in depth of ca. 10 m. This seems to be a lot compared to the fluctuations of 10-20 cm observed in the logger data (Fig. 1 in the reply of the authors).*

*Thus, larger changes in depth may be related to long-term processes (centennial to millennial scale) and minor changes to short-term events. This should be made clear in the text even if the logger data are not shown.*

Response: Right, so far we only recorded recharge-driven halocline fluctuations of 10–20 cm. However, there was no major recharge event since we started logging. It is likely that a heavy recharge event will lead to significantly higher fluctuations of the halocline.

We accounted for the referee´s concerns by altering the respective text passage in lines 11–12 on page 28 to "This range could solely depend on the hydraulic conditions, e.g. Hells Bells formation reflecting maximum and minimum elevations of the halocline as a result of droughts, recharge events and long-term sea level changes." With these changes and the changes that we introduced following the comments of anonymous referee #1, the matter of halocline elevation is now presented in a more general way in the revised manuscript. This means that instead of focusing on the recharge-driven halocline elevation as it was the case in the original manuscript, several causes for a halocline elevation on variable time-scales are now always discussed in the revised manuscript.

*In summary, I recommend to accept the MS after the suggested changes have been made.*

All changes to the revised manuscript can be tracked in the marked up version of the revised manuscript.

[revised manuscript text omitted]